Returning to the roots: resolution, reproducibility, and robusticity in the phylogenetic inference of Dissorophidae (Amphibia: Temnospondyli)

http://orcid.org/0000-0003-4517-3290 Gee Bryan M. bmgee@uw.edu
Burke Museum and Department of Biology, University of Washington , Seattle, WA , United States of America
Hutchinson John
Electronic publication date: 2021 Nov 8
Publication date: 2021
Volume: 9
Electronic Location ID: e12423
Received 2021 Mar 24; Accepted 2021 Oct 11
Copyright: © 2021 Gee
Copyright year: 2021
Copyright holder: Gee
License: This is an open access article distributed under the terms of the Creative Commons Attribution License, which permits unrestricted use, distribution, reproduction and adaptation in any medium and for any purpose provided that it is properly attributed. For attribution, the original author(s), title, publication source (PeerJ) and either DOI or URL of the article must be cited.
License URL: https://creativecommons.org/licenses/by/4.0/

Keywords: Olsoniformes, Dissorophoidea, Permian, Phylogeny, Paleozoic

Funding: NSF ANT-1947094 (to Chris Sidor) My current postdoctoral fellowship, under which I conducted this study, is supported by NSF ANT-1947094 (to Chris Sidor). The funders had no role in study design, data collection and analysis, decision to publish, or preparation of the manuscript.

==============================
The phylogenetic relationships of most Paleozoic tetrapod clades remain poorly resolved, which is variably attributed to a lack of study, the limitations of inference from phenotypic data, and constant revision of best practices. While refinement of phylogenetic methods continues to be important, any phylogenetic analysis is inherently constrained by the underlying dataset that it analyzes. Therefore, it becomes equally important to assess the accuracy of these datasets, especially when a select few are repeatedly propagated. While repeat analyses of these datasets may appear to constitute a working consensus, they are not in fact independent, and it becomes especially important to evaluate the accuracy of these datasets in order to assess whether a seeming consensus is robust. Here I address the phylogeny of the Dissorophidae, a speciose clade of Paleozoic temnospondyls. This group is an ideal case study among temnospondyls for exploring phylogenetic methods and datasets because it has been extensively studied (eight phylogenetic studies to date) but with most (six studies) using a single matrix that has been propagated with very little modification. In spite of the conserved nature of the matrix, dissorophid studies have produced anything but a conserved topology. Therefore, I analyzed an independently designed matrix, which recovered less resolution and some disparate nodes compared to previous studies. In order to reconcile these differences, I carefully examined previous matrices and analyses. While some differences are a matter of personal preference (e.g., analytical software), others relate to discrepancies with respect to what are currently considered as best practices. The most concerning discovery was the identification of pervasive dubious scorings that extend back to the origins of the widely propagated matrix. These include scores for skeletal features that are entirely unknown in a given taxon (e.g., postcrania in Cacops woehri) and characters for which there appear to be unstated working assumptions to scoring that are incompatible with the character definitions (e.g., scoring of taxa with incomplete skulls for characters based on skull length). Correction of these scores and other pervasive errors recovered a distinctly less resolved topology than previous studies, more in agreement with my own matrix. This suggests that previous analyses may have been compromised, and that the only real consensus of dissorophid phylogeny is the lack of one.

Introduction

Inferring phylogenetic relationships remains one of the most timeless pursuits within paleontology. The persistence of such studies owes to: (1) the great importance of phylogeny as the macroevolutionary framework within which all other studies are situated; and (2) the probable lability of any given topology when it relies entirely on morphological data and the discretization of continuous traits. While there is broad interest in exploring clade-independent practices that are applicable to a broad range of studies (e.g., comparison of likelihood and parsimony methods, approaches to missing data and polymorphisms), it is equally important to critically examine empirical datasets in order to assess their robusticity and reproducibility and to seek to improve them whenever possible.

One longstanding practice is the propagation of an existing matrix with modifications, at minimum by adding taxa of interest, and possibly more substantially by changing scores and character sampling. In this, there is an implicit goal of developing a semblance of a consensus matrix that the majority of workers have worked with (not necessarily within the confines of a single collaboration) and therein agree (or assume) is well-designed to test the relationships of a given in-group. However, in propagating a matrix, each derivate is inherently a pseudoreplicate (non-independent), especially when changes are minimal beyond taxon addition. While propagation creates consistency between analyses, it also constrains the possible outcomes unless substantial changes are made. If a consensus emerges from such a matrix, it must be assessed whether this is a truly defensible consensus. The easiest way to test this is with a novel (independent) matrix, assuming that the set of characters and their scoring do not substantially overlap with (converge on) those of previous matrices. Conversely, if no consensus emerges from repeated propagation of a largely unchanged matrix, this is perhaps even more troubling, as it indicates that most topologies are labile and thus should not be relied upon heavily for qualitative discussion or for integration into other studies.

This study focuses on the phylogenetic relationships of Dissorophidae, a clade of dissorophoid temnospondyls (Fig. 1), as a case study in phylogenetic analyses of temnospondyls. Temnospondyli, often referred to as ‘amphibians’ (nonamniote tetrapods in a broad historical sense and as the putative amphibian stem-group in more recent works), is best known for the clade’s longstanding role in the unresolved debate over lissamphibian origins. This ongoing debate has drawn extensive attention in recent decades (e.g., Laurin & Reisz, 1997; Anderson, 2001; McGowan, 2002; Ruta, Coates & Quicke, 2003; Vallin & Laurin, 2004; Lee & Anderson, 2006; Carroll, 2007; Ruta & Coates, 2007; Anderson et al., 2008a; Pyron, 2011; Sigurdsen & Green, 2011; Pardo, Small & Huttenlocker, 2017; Marjanović & Laurin, 2019; Daza et al., 2020; Schoch, Werneburg & Voigt, 2020). Workers interested in this topic have largely focused on Amphibamiformes, another dissorophoid clade, as the likely candidate within Temnospondyli. However, outside of this context, the phylogeny of most temnospondyl clades has received scant attention. Nonetheless, the intrarelationships of temnospondyls are of inherent import for this debate, especially with the recent proposal of a diphyletic origin of Lissamphibia from within Temnospondyli (Pardo, Small & Huttenlocker, 2017). Furthermore, with the increasing tractability of so-called “big data” studies with wide taxonomic breadth that are rooted in phylogenetic backbones, the phylogenetic relationships of these clades have acquired new import beyond the narrow scope of taxonomic specialists. Temnospondyls are both an excellent case study and an area in need of redress because they are not regarded as ‘charismatic taxa’ and thus suffer from a paucity of workers. As a result, certain workers or working groups may inadvertently exert disproportionate influence on the study of a given clade, which, in phylogenetics, manifests as one worker’s matrix rapidly becoming the only utilized matrix.

Figure 1 Cranial reconstructions of select representatives of Olsoniformes.

(A) The cacopine dissorophid Cacops morrisi (after Reisz, Schoch & Anderson, 2009); (B) The dissorophine dissorophid Dissorophus multicinctus (after Schoch, 2012); (C) The long-snouted trematopid Acheloma cumminsi (after Dilkes & Reisz, 1987; Polley & Reisz, 2011); (D) The short-snouted trematopid Ecolsonia cutlerensis (after Berman, Reisz & Eberth, 1985). Cool colors represent skull roof elements; warm colors represent palatal elements. Not to scale.

Dissorophids, a clade of dissorophoids only peripherally related to the lissamphibian origins debate, are an ideal case study within Temnospondyli. Firstly, the clade is very speciose, with over 20 nominal species. The anatomical foundation is well-established for most of these taxa, with nearly 20 studies published in the 21st century alone. As a result, there is a more substantive history of phylogenetic inquiry into dissorophids than for less speciose Paleozoic clades, such as eryopids or zatracheids. To date, there have been eight studies that addressed the phylogeny of Dissorophidae (Fröbisch & Reisz, 2012; Schoch, 2012; Holmes, Berman & Anderson, 2013; Maddin et al., 2013; Schoch & Sues, 2013; Liu, 2018; Dilkes, 2020; Gee et al., 2021). However, nearly all of them derive from the same source matrix (Schoch, 2012; Fig. 2) and are nearly identical in scoring, taxon sampling, and character sampling. Surprisingly, there is widespread disparity between their recovered topologies (Figs. 3, 4). As remarked upon by Dilkes (2020:26), “results of recent attempts to unravel the phylogeny of dissorophids, even though they are using modified and hopefully updated versions of the same matrix, have consistently shown a lack of agreement on a broader pattern of dissorophid relationships with poor resolution and low support for most nodes that are present in a strict consensus tree.” The situation of dissorophids can be characterized as a largely consistent matrix producing largely inconsistent results. Finally, the study of dissorophids is relevant for other studies. Dissorophids are a common outgroup for amphibamiform studies, whether in comparative anatomical descriptions or in quantitative studies (e.g., Pérez-Ben, Schoch & Báez, 2018; Atkins, Reisz & Maddin, 2019). Additionally, as ubiquitous components of Early Permian terrestrial ecosystems, dissorophids (and their sister group, Trematopidae), are frequently sampled in broader studies of Paleozoic tetrapods at large (e.g., Brocklehurst et al., 2018; Dunne et al., 2018; Pardo et al., 2019).

Figure 2 Genealogy of olsoniform-focused phylogenetic matrices.

Note that sources only reflect major contributions to character sampling. Number of characters is listed on the left, and number of all sampled taxa is listed on the right. ‘Focal clade’ refers to the most exclusive clade to which at least half of the sampled taxa belong. ‘Amphibamidae’ here refers to the historical concept of what is now Amphibamiformes (in part). Abbreviations: ARM, Atkins, Reisz & Maddin (2019); BHBK, Berman et al. (2010); BHMSA, Berman et al. (2011); D, Dilkes (2020); FR, Fröbisch & Reisz (2008, 2012); FS, Fröbisch & Schoch (2009); G, Gee (2020b); G* (2021), this study; GBHPH, Gee et al. (2021); GR, Gee & Reisz (2019); HBA, Holmes, Berman & Anderson (2013); L, Liu (2018); MFEM, Maddin et al. (2013); PR, Polley & Reisz (2011); RB, Ruta & Bolt (2006); S, Schoch (2012, 2018a); SHH, Schoch, Henrici & Hook (2020); SM, Schoch & Milner (2008, 2021); SR, Schoch & Rubidge (2005); SS, Schoch & Sues (2013); SW, Schoch & Witzmann (2018).

Figure 3 Comparison of tree topologies from early phylogenetic analyses of Dissorophidae.

(A–D) All topologies represent strict consensus trees except for Schoch (2012) and are visually truncated to depict only dissorophids. Cacopinae and Dissorophinae are not annotated for Fröbisch & Reisz’s topology because the taxonomic specifiers (Cacops aspidephorus and Dissorophus multicinctus) were not sampled. Conjunctio multidens represents a composite OTU unless otherwise indicated by the differentiation of the holotype from the specimen historically referred to as the Rio Arriba Taxon (RAT; UCMP 40103). Nominal placement and nodal definitions from Schoch & Milner (2014).

Figure 4 Comparison of reported tree topologies from recent phylogenetic analyses of Dissorophidae.

(A–D) All topologies represent strict consensus trees and are visually truncated to depict only dissorophids. Colors and symbols as with Fig. 3. Conjunctio multidens represents a composite OTU unless otherwise indicated; Gee et al. (2021) recovered all three specimens as a clade, so they are collapsed to a single visual OTU here. For Holmes, Berman & Anderson (2013), the tree on the left represents the result of their analysis with scoring changes to the first referred specimen of C. multidens (UCMP 40103 (RAT)); the tree on the right represents the result with the original scorings from Schoch (2012). For Dilkes (2020), the tree on the left represents the result of his analysis with the full character and taxon sample; the tree on the right represents the result following the removal of wildcard taxa. Nominal placement and nodal definitions from Schoch & Milner (2014).

I previously addressed the phylogeny of Trematopidae, the sister group to Dissorophidae (collectively Olsoniformes; Anderson et al., 2008b), which is less studied and for which there also remains no consensus (Gee, 2020b; Fig. 5). A key aspect of my previous study was demonstrating how taxon sampling (and more specifically, selective exclusion) can drastically improve or alter topological resolution. Previous olsoniform studies have either focused on dissorophids or trematopids–there has never been a matrix that evenly samples these clades as a collective in-group. The closest approximation is Atkins, Reisz & Maddin (2019), a derivate of Schoch (2018a) that samples six of the 10 trematopids (deficient in Carboniferous taxa as with other studies) and 11 dissorophids. Schoch’s (2018a) dissorophoid matrix is in turn derived from Schoch’s (2012) dissorophid matrix, though with a different taxon sample than that of Atkins, Reisz & Maddin.

Figure 5 Comparison of tree topologies from previous phylogenetic analyses of Trematopidae.

(A–D) All topologies represent strict consensus trees and are visually truncated to depict only xerodromes. Nominal placement and nodal definitions from Schoch & Milner (2014).

In this study, I expand the taxon and character sampling of my previous trematopid-centric matrix to encompass dissorophids, thereby forming such a matrix. I opted to use my matrix as the foundation, rather than the long-propagated dissorophid matrix of Schoch (2012) or more modified derivates like that of Schoch (2018a). This decision was motivated by a desire to approach these questions from as independent of a perspective as possible and in light of the markedly disparate topologies of different derivates of Schoch’s matrix despite the matrix’s conserved nature. While it broadly samples Olsoniformes, this study is targeted primarily towards dissorophids simply because I have recently focused on trematopids. There are three primary objectives: (1) to test whether this independent matrix can produce either better resolution or stronger nodal support compared to previous studies and in turn to interpret that resolution; (2) to assess factors that might contribute to potentially spurious resolution in analyses that recover a high degree of resolution or that confound recovery of resolution in analyses that recover a low degree of resolution; and (3) to summarize the present state of Dissorophidae and to identify key areas in need of redress in order to work towards a consensus.

Materials & Methods

Taxon sampling

I sampled the vast majority of olsoniforms, with all 10 of the undisputed trematopid species carried over from my previous analysis (Gee, 2020b). Acheloma dunni is regarded as a junior synonym of Acheloma cumminsi, and their scores are thus merged here. Phonerpeton whitei is excluded on the basis of a suspect differentiation from Phonerpeton pricei, which stems from the absence of detailed description or illustration. I expanded the sample to include most dissorophids, the recently described olsoniform Palodromeus bairdi (Schoch, Henrici & Hook, 2020), and the putative ‘basal’ dissorophoid Perryella olsoni (Table 1).

Table 1 Summary of newly added olsoniform taxa.

Taxon	Time	Location	References	Completeness	
Anakamacops petrolicus	Middle Permian	China (Gansu)	Li & Cheng (1999), Liu (2018)	68 (62.3%)	
Aspidosaurus binasser	Early Permian	USA (TX)	Berman & Lucas (2003)	55 (50.4%)	
Aspidosaurus chiton	Early Permian	USA (TX)	Broili (1904)	22 (20.1%)	
Aspidosaurus novomexicanus	Late Carboniferous	USA (NM)	Williston (1911), Carroll (1964a)	23 (21.1%)	
Brevidorsum profundum	Early Permian	USA (TX)	Carroll (1964a)	30 (27.5%)	
Broiliellus arroyoensis	Early Permian	USA (TX)	DeMar (1967)	25 (22.9%)	
Broiliellus brevis	Early Permian	USA (TX)	Carroll (1964a)	79 (72.4%)	
“Broiliellus” hektotopos	Early Permian	USA (OH)	Berman & Berman (1975)	40 (36.6%)	
Broiliellus olsoni	Early Permian	USA (TX)	DeMar (1967), Bolt (1974b)	37 (33.9%)	
Broiliellus reiszi	Early Permian	USA (NM)	Holmes, Berman & Anderson (2013)	88 (80.7%)	
Broiliellus texensis	Early Permian	USA (TX)	DeMar (1966b), Bolt (1974b)	61 (55.9%)	
Cacops aspidephorus	Early Permian	USA (TX)	Williston (1910), Anderson (2005), Dilkes & Brown (2007), Dilkes (2009), Anderson, Scott & Reisz (2020)	103 (94.5%)	
Cacops morrisi	Early Permian	USA (OK)	Reisz, Schoch & Anderson (2009), Gee & Reisz (2018a), Gee, Bevitt & Reisz (2019)	98 (89.9%)	
Cacops woehri	Early Permian	USA (OK)	Fröbisch & Reisz (2012), Fröbisch, Brar & Reisz, 2015, Gee, Bevitt & Reisz (2019)	72 (66.0%)	
Conjunctio multidens	Early Permian	USA (CO, NM)	Case & Williston (1913), Carroll (1964a), Schoch & Sues (2013), Gee et al. (2021)	59 (54.1%)	
Diploseira angusta	Early Permian	USA (TX)	Dilkes (2020)	50 (45.8%)	
Dissorophus multicinctus	Early Permian	USA (TX)	DeMar (1968); Milner (2003); Dilkes (2020)	101 (92.6%)	
Iratusaurus vorax	Middle Permian	Russia (Bashkortostan)	Gubin (1980)	12 (11.0%)	
Kamacops acervalis	Middle Permian	Russia (Perm Krai)	Gubin (1980); Schoch (1999)	31 (28.4%); 51 (46.7%)	
Nooxobeia gracilis	Middle Permian	USA (OK)	Gee, Scott & Reisz (2018)	19 (17.4%)	
Palodromeus bairdi	Late Carboniferous	USA (OH)	Schoch, Henrici & Hook (2020)	70 (62.5%)	
Parioxys bolli	Early Permian	USA (TX)	Carroll (1964b)	11 (10.0%)	
Platyhystrix rugosa	Early Permian	USA (CO, NM)	Berman, Reisz & Fracasso (1981)	67 (61.4%)	
Reiszerpton renascentis	Early Permian	USA (TX)	Maddin et al. (2013)	56 (51.3%)	
Scapanops neglectus	Early Permian	USA (TX)	Carroll (1964a), Schoch & Sues (2013)	57 (52.2%)	
Zygosaurus lucius	Early Permian	Russia (Bashkortostan)	Eichwald (1848), Efremov (1937)	19 (17.4%)	
Note:

Completeness refers to percent of characters that could be scored; note that this includes cells scored as inapplicable (-), even though most programs treat these as missing data (?).

In my previous study, I scored practically every published and figured trematopid specimen. I opted for a more selective approach in adding dissorophids, and the trematopid specimen-level OTUs are not carried over. The main reason is that many dissorophid specimens consist only of neural spines and osteoderms (e.g., DeMar, 1966b; May et al., 2011; Gee, Bevitt & Reisz, 2019) and can only be scored for a handful of characters (<10%). Dissorophid taxa that are excluded in their entirety are: Aspidosaurus glascocki, “Aspidosaurus” apicalis, “Aspidosaurus” crucifer, “Aspidosaurus” peltatus, and Astreptorhachis ohioensis. Very fragmentary cranial remains or those without sutures were excluded unless they were the holotype of a valid taxon (e.g., Broiliellus arroyoensis), as similar specimens were frequently problematic in my trematopid analysis. This exclusion includes the holotypes of “Fayella chickashaensis” and “Trematopsis seltini,” both too poorly preserved to be scored, and two junior synonyms of D. multicinctus, “Otocoelus mimeticus” and “Otocoelus testudineus,” both characterized only by relatively brief and dated descriptions (Cope, 1896a, 1896b). The only holotype of a junior synonym that is sampled here is “Longiscitula houghae” (=D. multicinctus) because its cranial sutures are well-described and figured (DeMar, 1966a; Milner, 2003). Additionally, testing interspecific ontogenetic disparity in dissorophids was not a focus of this study (addressed in the Discussion), and therein assessing whether the matrix could detect intraspecific ontogenetic variation as a proof of concept was not as essential. Almost two-thirds of dissorophids are represented only by the holotype in any event. I did run one analysis to assess this, and thus Anakamacops petrolicus, Cacops morrisi, Cacops woehri, and Conjunctio multidens are scored at the specimen level.

Schoch & Milner (2014) listed several taxa as possible early-diverging dissorophoids: Macrerpeton huxleyi from the Late Carboniferous of Ohio; Parioxys ferricolus and Parioxys bolli from the Early Permian of Texas; Perryella olsoni from the Early Permian of Oklahoma; and Stegops newberryi from the Late Carboniferous of Ohio. Parioxys ferricolus and S. newberryi are in need of redescription. Parioxys bolli (Carroll, 1964b) is included to test the performance of an exclusively postcranial skeleton (the holotype and only known specimen). Perryella olsoni is included, but its position should only be interpreted in light of assumed dissorophoid affinities. Ruta & Bolt (2006) considered it to be a dvinosaur and recovered it as such in a broad analysis of Paleozoic temnospondyls, while Schoch (2018a) recovered it as the sister taxon to Dissorophoidea but in a dissorophoid-focused analysis with only one dvinosaur taxon. Schoch & Milner (2021) recently recovered M. huxleyi as the sister-group of all other dissorophoids (what they term a “stem dissorophoid”) diverging before P. olsoni, so the former is not sampled here. “Broiliellus” hektotopos from the Early Permian of Ohio was described as a dissorophid (Berman & Berman, 1975), but it has never been included in an analysis and is probably an amphibamiform (May et al., 2011), following Romer’s (1952) initial interpretation of the holotype as a “branchiosaur.” I included the holotype to test this in a phylogenetic framework.

The non-olsoniform outgroups were carried over from Gee (2020b): the amphibamiforms Doleserpeton annectens, Eoscopus lockardi, Pasawioops mayi, and Tersomius texensis, and the micromelerpetid Micromelerpeton credneri. The same non-dissorophids were included (Dendrysekos helogenes, Eryops megacephalus) but with Chenoprosopus milleri (Langston, 1953) replacing Chenoprosopus lewisi to better sample palatal characters. I replaced my previous functional outgroup, the anthracosaur Proterogyrinus scheelei, with the colosteid Greererpeton burkemorani (Smithson, 1982; Godfrey, 1989a, 1989b; Bolt & Lombard, 2001) because the latter’s preservation allows it to be confidently scored for more characters without relying on reconstructions (e.g., skull length-based characters), but scores for P. scheelei were still updated.

Character sampling

The character matrix was derived from that used in my trematopid analysis (Gee, 2020b), and characters are listed in Appendix 1. Additional characters were added to differentiate between dissorophids. I removed five characters (Appendix 2) and split one character (lateral exposure of the palatine; LEP) into two characters. There are a total of 109 characters; multi-state characters that can be hypothesized to be ordered in an evolutionary sense were ordered here. The decision to order certain characters follows other workers (e.g., Marjanović & Laurin, 2019), including other dissorophid studies (e.g., Dilkes, 2020), and is motivated by two factors: (1) that leaving characters unordered is neither neutral nor a lack of assumption, as unordered characters imply that transitions between all states are equally likely (e.g., Slowinski, 1993; Wiens, 2001); and (2) that various studies have shown that ordering characters tends to improve the ability to recover genuine clades and to increase topological resolution (e.g., Fröbisch & Schoch, 2009; Grand et al., 2013; Rineau et al., 2015; Simões et al., 2017; Rineau, Zaragüeta i Bagils & Laurin, 2018). Of the 18 multistate characters, 15 were ordered (2, 4, 11, 18, 20, 28, 31, 38, 51, 68, 70, 84, 99, 104, 107); characters 19, 88, and 98 are left unordered. Characters were equally weighted.

Character scoring

Scores were carried over at the species level from Gee (2020b) unless noted in Appendix 2. Scoring was based strictly on the literature, and reconstructions were not utilized. Scoring of species-level operational taxonomic units (OTU) accounted for any polymorphisms that are not clearly attributable to ontogeny. Anatomical differences correlated with very slight size differences were treated more skeptically than those correlated with large size gaps in determining whether to code a condition as polymorphic or only for the inferred ‘adult’ condition. Comments on scoring approaches to specific characters and the treatment of taxa for which a character can only be scored from an immature specimen are provided in Appendix S2 of Gee (2020a) and in Appendix 3 here. The matrix was compiled in Mesquite version 3.6b917 (Maddison & Maddison, 2020) and is provided as a NEXUS file in Appendix 4.

I want to make particular mention of Kamacops acervalis here because it is difficult to reconcile its historical scoring with existing descriptions, figures of which are either reconstructions in palatal and lateral view or close-up illustrations of the occiput and braincase (Gubin, 1980; Schoch, 1999). The detailed discussion is relegated to Appendix 3, but I emphasize a few key points here because this taxon in particular (or rather, uncertainty over the taxon) influenced the design of the analyses that I conducted here.

The only published photograph of any specimen is a snout (Schoch & Milner, 2014:fig. 37E), which does not show any clear sutures. This taxon has also been reconstructed as a silhouette in dorsal view but without sutures by Schoch (2012:fig. 6), in contrast to most other dissorophids. Therefore, it is strange that there are cranial characters scored for this taxon (e.g., prefrontal-postfrontal suture). Additionally, none of the specimens of Kamacops acervalis is even half-complete, and without photographs, it is unclear how much skeletal overlap exists between specimens, therein questioning the fidelity of the reconstructions. This is particularly important because the reconstruction is the only means of scoring characters involving skull length, including two of the three characters that differentiate K. acervalis from Cacops. At present, many of the historic scores cannot be reconciled with the literature. Schoch indicated that he personally examined material of this taxon, so it is possible that he identified new features, but the data to support the scoring of many characters are not established in the literature. Therefore, I constructed two different species-level OTUs for this taxon. The first is termed the ‘conservative’ OTU and scores only based on the explicit descriptions and specimen illustrations (but not reconstructions) of Gubin (1980) and Schoch (1999). The second is termed the ‘reconstructed’ OTU and augments the previous one with data from reconstructions; this led to the scoring of an additional 20 characters. In neither OTU are skull roof sutures scored except in the occipital region, so both OTUs are underscored compared to previous studies.

Phylogenetic analysis

I elected to perform only maximum parsimony analyses of the data matrix; the topologies recovered by a simple non-clock Bayesian analysis in my previous study (Gee, 2020b) did not differ substantially from those recovered by the parsimony analyses. Most discrepancies were related to additional resolution recovered in the Bayesian analysis only on account of the overestimation of support of posterior probabilities compared to bootstrap value (e.g., Alfaro, Zoller & Lutzoni, 2003; Cummings et al., 2003; Douady et al., 2003; Erixon et al., 2003; Simmons, Pickett & Miya, 2004; Zander, 2004).

The analyses were primarily performed in TNT v1.5 (Goloboff & Catalano, 2016). For analyses with more than 30 taxa, I used a heuristic search (“traditional search” in TNT) with the following parameters: 10,000 random addition sequence replicates, holding 10 trees at each step, and tree-bisection-and-reconnection (TBR). All sets of MPTs were then used as the starting trees for a second round of branch swapping to obtain the final set of MPTs. For analyses with 30 or fewer taxa, a branch-and-bound (“implicit enumeration” in TNT) search was used. Default settings of TNT (e.g., rule 1 for branch collapsing: min. length = 0) were otherwise maintained. I performed 10,000 bootstrap replicates with a heuristic search to assess absolute nodal support for all analyses. Given the intensive computation time of branch-and-bound searches, I elected to use heuristic searches to calculate Bremer decay indices after confirming that a heuristic search recovered the same set of MPTs as the branch-and-bound search. TNT was selected for its ability to rapidly process large datasets, which was a concern for analyses with species-level OTUs that have high proportions of missing data (as with some of the dissorophids here); no search (or resampling) exceeded 9 h. I used PAUP* 4.0a169 (Swofford, 2021) for several analyses that mirrored previous analyses conducted in PAUP*. The more taxonomically restricted TNT analyses can be run within PAUP* (no search exceeded 36 h), although these analyses were consistently more costly, and bootstrapping could exceed 48 h. One noteworthy difference is that to the best of my knowledge (based on the associated publications and other online resources), TNT cannot handle partial uncertainty (or at least the syntax used for this in NEXUS files is incompatible). These scores were thus changed to polymorphisms encompassing the possible character states in TNT analyses (e.g., a partial uncertainty score of ‘{0 1}’ was scored as ‘(0 1)’ in TNT; this is how Mesquite treats partial uncertainty when exporting a NEXUS file to TNT). These are computationally the same but with an additional step added to the tree for each polymorphism compared to a partial uncertainty. All analyses were performed on a personal computer (MacBook Pro, 2015 model, 16 GB of RAM, macOSMojave 10.14.5).

As with my previous study, I performed a large number of analyses (Table 2). Almost all of the trematopid analyses were determined a priori, but because the focus of this study was different, most of these analyses were only determined after examining previous results. The first four analyses broadly sample Olsoniformes and have an eye towards testing topology using standard historical methods (e.g., sampling all taxa, removal of wildcards), whereas the last five analyses relate more to assessing possible sources of disparity between previous studies.

Table 2 Summary of the permutations performed in this study, indicating software, search type, and general characterization of the analysis with respect to taxon and character sampling.

Analysis	Software	Search	Taxon sample	Character matrix	
1A	TNT	Heuristic	This study	This study	
1B	TNT	Heuristic	This study	This study	
2	TNT	Heuristic	This study	This study	
3	TNT	Heuristic	This study	This study	
4	TNT	Heuristic	This study	This study	
5	PAUP*	Both	Dilkes (2020)	This study	
6	TNT	Branch-and-bound	Gee (2020b)	This study	
7	PAUP*	Branch-and-bound	Dilkes (2020)	Dilkes (2020), with scoring changes	
8	PAUP*	Branch-and-bound	Gee (2020b)	This study	
9A	TNT	Branch-and-bound	Dilkes (2020)	Dilkes (2020), original scores	
9B	TNT	Branch-and-bound	Dilkes (2020)	Dilkes (2020), with scoring changes	
Note:

The heuristic search in TNT is termed ‘traditional search,’ and the branch-and-bound search is termed ‘implicit enumeration.’

1. Analysis 1A (all olsoniforms): all presently valid species that I scored are sampled here at the species level. Taxa: 47.

2. Analysis 1B (wildcard removal): an Adams consensus was used to identify wildcard taxa in the previous analysis (method for identifying wildcards is listed in the Results). The analysis was subsequently rerun without these wildcards. Taxa: 30/33 (number differs due to different wildcard identification).

3. Analysis 2 (best representatives): this analysis follows historic approaches by excluding poorly known taxa, which are arbitrarily defined as having either an overall low percentage of scoreable features or a low percentage specifically for cranial characters. The sampling thus omits any taxon for which cranial sutures are unknown: Aspidosaurus chiton, Broiliellus arroyoensis, Iratusaurus vorax, Parioxys bolli, and Zygosaurus lucius. Cranial material of Aspidosaurus novomexicanus, Brevidorsum profundum, Diploseira angusta, Kamacops acervalis, and Nooxobeia gracilis is relatively fragmentary, and these taxa are also excluded. Of the exclusions, B. profundum, K. acervalis, and Z. lucius were usually sampled in previous studies. All non-dissorophids are sufficiently characterized to be retained. Taxa: 37.

4. Analysis 3 (dissorophid-focused): this analysis samples almost every nominal dissorophid but with a trimmed subset of trematopids, as would commonly be done for dissorophid-focused analyses (e.g., Schoch, 2012, and derivates thereof). I excluded Actiobates peabodyi, Mordex calliprepes, and Rotaryus gothae as taxa probably represented only by markedly immature specimens (these are the smallest trematopids). “Broiliellus” hektotopos and Parioxys bolli were excluded since they were recovered well outside Dissorophidae in previous analyses. Taxa: 42.

I ran one analysis with specimen-level dissorophid OTUs to assess whether the matrix can resolve intraspecific ontogeny in dissorophids. 5. Analysis 4 (cacopine specimen-level OTU): this analysis focuses on Anakamacops petrolicus, the three species of Cacops, and Conjunctio multidens, for which at least two specimens can be scored. Based on my trematopid analyses, poorly preserved or highly fragmentary specimens were clear confounds. Therefore, I omitted highly fragmentary referred material (like the two partial snouts referred to Ca. woehri by Gee, Bevitt & Reisz, 2019; ROMVP 80800, ROMVP 80801) and material without sutures (like the holotype and paratype of Ca. aspidephorus; FMNH UC 647, FMNH UC 649). Iratusaurus vorax and Zygosaurus lucius are entirely excluded on the same grounds. Species-level dissorophid OTUs are restricted to Broiliellus brevis, Dissorophus multicinctus and Kamacops acervalis. Non-olsoniforms include Chenoprosopus milleri, Dendrysekos helogenes, Doleserpeton annectens, Eoscopus lockardi, and Eryops megacephalus, and the outgroup, Greererpeton burkemorani. OTUs: 32.

The third set of analyses relate to taxon sampling. As with my previous study, I ran so-called “mirror analyses” in which I took the taxon sample of a previous study and the character sample and scoring of my own study. 6. Analysis 5 (taxon mirror of Dilkes (2020)): this analysis mirrored the taxon sample of Dilkes (2020). Because not all equivalent taxa are found in my matrix, I replaced Sclerocephalus haeuseri with Eryops megacephalus and Platyrhinops lyelli with Eoscopus lockardi. Dendrysekos helogenes was utilized as the outgroup. I used the ‘reconstructed’ OTU of Kamacops acervalis to approximate the historic scoring of this taxon. Although I intended to mirror all of Dilkes’ parameters in PAUP*, a preliminary run with a branch-and-bound search produced no progress after 24 h. This is not surprising as 29 taxa are already above the typical threshold for running this search. Therefore, I ran a heuristic search with 10,000 random addition sequence replicates, holding 10 trees per step; all other settings, including the bootstrapping, were mirrored where possible. I then removed the four wildcard taxa that Dilkes identified and reran the analysis; this trimmed sample was recognized to be tractable with a branch-and-bound search. Taxa: 29 (25 without wildcards).

7. Analysis 6 (taxon mirror of Gee (2020b)): this analysis mirrored the taxon sample of Gee (2020b). Acheloma and Phonerpeton are treated as discussed above. I used Proterogyrinus scheelei as the outgroup (as with the original study) and excluded Greererpeton burkemorani. In order to account for the adjusted scoring of A. cumminsi and the exclusion of Ph. whitei, I reanalyzed my original matrix with the revised treatments of these taxa but with the original scoring otherwise intact. Since only one OTU was changed, the revised string for Acheloma cumminsi (inclusive of data from Acheloma dunni) is provided in Appendix 2 rather than in a separate NEXUS file. Taxa: 23.

The next analysis resulted from a close examination of previous scores of the Schoch (2012) matrix and its derivates in which I identified a large number of cells that either were scored for features that are definitively unknown (e.g., postcrania of Cacops woehri) or that were unscored but that are definitely known (e.g., palatal features of Fedexia striegeli). Most of the cells that were spuriously scored are scored identically to other members of a presumed close relative by Schoch (e.g., questionable scores of Cacops aspidephorus were scored identical to those of Cacops morrisi), although some were introduced by Holmes, Berman & Anderson (2013); (e.g., all questionable scores of C. woehri were scored identical to at least one other species of Cacops). This observation suggests that some scores have been “assumed,” but not actually observed, based on inferred relatedness, a troubling prospect. 8. Analysis 7 (updated version of Dilkes (2020)): this analysis is a direct reanalysis of Dilkes’ matrix with updated scores that focused on cells where the scoring or lack thereof seems unequivocally erroneous. The primary objective therein is to strictly examine the influence of dubious scorings on the matrix. I did not adjust scores unless there was strong evidence against the current score, so subjective decisions (e.g., is an atlas-axis sufficient postcranial representation to determine the absence of osteoderms in Fedexia striegeli) were not changed. I only rescored cells for polymorphisms when this condition was not clearly linked to size variation and was unequivocally non-taphonomic. One note is that the Acheloma of this matrix is specifically “Acheloma dunni”; I only updated scores based on material referred to this junior synonym (Maddin, Reisz & Anderson, 2010; Polley & Reisz, 2011). This differs from my own matrix in which Acheloma cumminsi is scored from both originally referred material and that of “A. dunni,” a junior synonym (Gee, 2020b). Changes were not made to taxon sampling, character sampling, or character construction (including ordering), even though certain characters of Dilkes’ matrix (e.g., palpebral ossifications) are intentionally excluded in my own matrix. These approaches minimize personal scoring philosophy and should provide an acceptable derivate of this matrix should other workers continue to use it. Any restored scores will have to be justified with appropriate data. All scoring changes are listed and justified in Appendix 5, and the revised matrix is provided as Appendix 6. The analysis was rerun in PAUP* following Dilkes’ parameters. Note that I first analyzed Dilkes’ original matrix in PAUP* as well to ensure that my program settings recovered the same results.

Finally, when considering explanations for disparity in the degree of resolution and the placement of wildcard taxa, I observed that different programs have been used to examine the Schoch (2012) matrix and its derivates. Schoch (2012), Holmes, Berman & Anderson (2013), Maddin et al. (2013), and Dilkes (2020) used PAUP*, whereas Schoch & Sues (2013) and Liu (2018) used TNT. The analyses using TNT have recovered more resolution in the strict consensus and without any wildcards. It is known that these programs’ algorithms and default settings differ, but studies rarely compare the results obtained by analyzing the same matrix. Therefore, I performed two analyses that analyze the same matrix with each program; I term these “parallel analyses.” These analyses are not intended to test whether one program’s settings and results can be reproduced in the other but rather to test whether default parameters and algorithmic differences (usually default settings are employed by other workers) produce different results. 9. Analysis 8 (parallel of trematopid mirror): this analysis parallels Analysis 6 (trematopid-focused sample, analyzed using TNT) in PAUP*; this analysis was chosen because it achieved a measurable degree of resolution in TNT and has a low taxon sample that would make it tractable for a branch-and-bound search in PAUP*.

10. Analysis 9 (parallel of Dilkes (2020)): this analysis parallels the original results of Dilkes (2020) and the updated version of that matrix that I analyzed in Analysis 7, both using TNT (Dilkes ran his analysis in PAUP*). Dilkes ran three analyses with a branch-and-bound search: a full taxon sample, a full taxon sample with osteoderm characters removed, and a reduced taxon sample without wildcards but with the full character sample. I paralleled the two with the full character sample. The iterations comparing different programs’ analysis of Dilkes’ original matrix are termed Analysis 9A, and the iterations comparing the TNT analysis of the original and updated versions of Dilkes’ matrix are termed Analysis 9B.

Two nomenclatural notes

The dissorophid subfamily defined by all taxa more closely related to Cacops aspidephorus than to Dissorophus multicinctus has a complex history. Cacops was historically referred to Aspidosaurinae (e.g., Williston, 1914; DeMar, 1966b; Milner, 2003; Witzmann & Soler-Gijón, 2010). However, Daly (1994) considered the poorly known Aspidosaurus to be a dissorophine, making Aspidosaurinae a junior synonym of Dissorophinae and necessitating a new name for historical aspidosaurines that could not be placed in Dissorophinae (like Cacops); to this end, she coined the name ‘Cacopinae.’ However, Daly did not substantiate her classification, so Cacopinae was unused until the study of Schoch & Rubidge (2005), who used it as a terminal OTU (of Cacops aspidephorus + Kamacops acervalis); note that this study did not include Aspidosaurus. The use of Cacopinae was then followed by nearly all subsequent workers (except Witzmann & Soler-Gijón (2010)) but with the original source only noted by Berman et al. (2010). The name was then changed to Eucacopinae by Schoch & Sues (2013), who noted that Cacopinae, the proper derivation from Cacops, was preoccupied for a group of microhylid frogs named for “Cacopus” (=Uperodon) per Noble (1931); Cacopinae sensu Noble became a junior synonym of Microhylinae. The erection of Eucacopinae was considered to be the appropriate solution by Schoch and Sues. However, as has been brought to my attention by David Marjanović, there are two fundamental flaws with this nomenclatural act. The first is that per Article 11.7.1.1 of the International Code of Zoological Nomenclature (International Commission on Zoological Nomenclature (ICZN), 1999), family names ‘must be […] formed from the stem of an available generic name,’ but there is no genus bearing the name ‘Eucacops’ or something similar. Furthermore, Noble’s (1931) derivation of ‘Cacopinae’ from ‘Cacopus’ only includes part of the stem (which is ‘Cacopod,’ not ‘Cacop’), and, per the same article, is unavailable. Therefore, Cacopinae Daly, 1994, is both the valid derivation from Cacops and not preoccupied, so it need not be replaced. Schoch (2018a), Atkins, Reisz & Maddin (2019), Anderson, Scott & Reisz (2020), and Schoch & Milner (2021) recently employed Cacopinae, without comment, which I follow here. Cacopinae may yet prove to be a junior synonym of Aspidosaurinae, but this is not supported by the following results of this study.

Also as pointed out to me by David Marjanović, Platyhystrix is feminine (following the gender of ‘hystrix’), and therefore, when Ctenosaurus rugosus Case, 1910, was transferred to Platyhystrix Williston, 1911, nomenclatural standards dictated that it be changed to Platyhystrix rugosa (per Article 34.2 of the ICZN), even though this was not actually put into practice until this study, over a century later. This change is implemented throughout this manuscript.

Results

My reporting practices are outlined here to reduce redundancy. Figured topologies are either strict consensus trees or Adams consensus trees (only Analysis 1A). The associated MPTs of each analysis are included as .tre files in Appendix 7. All bootstrap values are reported, so any node without a listed bootstrap value was not recovered in more than 1% of the bootstrap replicates (for TNT analyses) or more than 5% of the replicates (for PAUP*). Note that bootstrap frequencies reported for TNT analyses are absolute frequencies, not frequency differences (GC), which are the program’s default. Because bootstrap support below 50% and Bremer support below three are not considered strong, any values below these thresholds are colored in grey, whereas any values at or above these thresholds are colored in black.

The node-based definition of Dissorophoidea (the least inclusive grouping with Dissorophus multicinctus and Micromelerpeton credneri) and the stem-based definitions for Trematopidae (most inclusive clade containing Acheloma cumminsi but not D. multicinctus), Dissorophidae (specifiers of Trematopidae inverted), Cacopinae (most inclusive clade containing Cacops aspidephorus but not D. multicinctus), and Dissorophinae (specifiers of Cacopinae inverted) are used here following Schoch & Milner (2014). Therefore, all of these clades are “recovered” in any analysis in which the specifiers are sampled, but they may include only that specifier (e.g., only A. cumminsi for Trematopidae). Some specifiers for the node-based definitions of Xerodromes, Amphibamiformes, and Olsoniformes are not sampled in this analysis (Amphibamus grandiceps and Apateon pedestris), but these names are used in the same sense as other workers (the sister group of Micromelerpetidae, the clade of all small-bodied xerodromes, and the sister group of Amphibamiformes, respectively). The looser definitions of these clades in this context means that they are not always recovered and must include at least two taxa that form a clade. Nominal placement of taxa, specifically for dissorophids, is also from Schoch & Milner (2014). These definitions are color-coded in the figures to visually facilitate the comparison of topologies and placement of taxa.

Analysis 1A (all olsoniforms)

The iteration with the ‘conservative’ OTU of Kamacops acervalis recovered 93,116 MPTs with a length of 401 steps (CI = 0.314; RI = 0.592). The strict consensus is predictably unresolved beyond a node for Temnospondyli and the default node excluding the operational outgroup (Greererpeton burkemorani). The iteration with the ‘reconstructed’ OTU of K. acervalis recovered 21,646 MPTs with the same length of 401 steps and an identical strict consensus topology. Because of the total lack of resolution, these topologies are not presented here.

Adams consensus trees were computed for both iterations in order to identify wildcard taxa (Fig. 6). The topologies are largely consistent with the main differences lying in the composition of Cacopinae. All nominal trematopids form a clade, but all nominal dissorophids do not. Parioxys bolli is recovered entirely outside of Dissorophoidea in a polytomy at the base of Temnospondyli; Iratusaurus vorax and Reiszerpeton renascentis are recovered in a polytomy outside of Olsoniformes with Palodromeus bairdi; and Platyhystrix rugosa is recovered in a polytomy with Trematopidae and Dissorophidae (as defined above). “Broiliellus” hektotopos is recovered in a polytomy with individual branches for all four amphibamiforms and a branch for all nominal dissorophids other than Par. bolli at the base of Xerodromes.

Figure 6 Adams consensus trees for Analysis 1A (all scored olsoniforms).

(A) Tree resulting from the iteration using the ‘conservative’ OTU of Kamacops acervalis; (B) tree resulting from the iteration using the ‘reconstructed’ OTU of K. acervalis. Only Dissorophidae is depicted in part B because the remainder of the consensus tree was unchanged from part A.

Wildcard identification was restricted to nominal olsoniforms. The highly incomplete Parioxys bolli was removed because it falls well outside of Olsoniformes in the Adams consensus. Palodromeus bairdi was also removed given its position outside of Olsoniformes. Iratusaurus vorax, Platyhystrix rugosa, and Reiszerpeton renascentis were removed given their position outside of Dissorophidae. Within Dissorophidae, I removed any nominal dissorophid that was part of a polytomy at the base of the clade; this led to the removal of Aspidosaurus chiton, Aspidosaurus novomexicanus, Brevidorsum profundum, and Nooxobeia gracilis, all of which are poorly known taxa. In the iteration with the ‘conservative’ OTU of Kamacops acervalis, this taxon was also recovered in this basal polytomy and was removed. Similar to Dilkes (2020), I then removed any taxon recovered in a polytomy with Cacopinae and Dissorophinae, leading to the removal of Aspidosaurus binasser and Scapanops neglectus in both iterations, as well as Anakamacops petrolicus and Zygosaurus lucius in the iteration with the ‘conservative’ OTU of K. acervalis. One notable retention in both iterations is Conjunctio multidens, a wildcard as identified by Dilkes (2020). The two iterations differ in dissorophid composition by three taxa. For Trematopidae, I removed any taxon that was part of a basal polytomy; the pair of Actiobates peabodyi + Mattauschia laticeps and Mordex calliprepes met this criterion in both iterations. These are the more fragmentary trematopids and have rarely been sampled in previous analyses.

Following the wildcard identifications, taxa were then removed from the original sets of MPTs, and the strict consensus was recalculated. When newly identical MPTs were subsequently removed, there was a substantial reduction in the number of unique MPTs. There are 122 remaining MPTs for the iteration with the ‘conservative’ OTU of Kamacops acervalis and 346 remaining MPTs for the iteration with the ‘reconstructed’ OTU. The strict consensus of the retained taxa of both iterations (Fig. 7) largely follows that observed in the Adams consensus (Fig. 6), though with a loss of resolution in Trematopidae and Dissorophinae. Micromelerpeton credneri and Perryella olsoni now form a polytomy with amphibamiforms and “Broiliellus” hektotopos instead of forming an earlier diverging branch (i.e., Xerodromes is not recovered). In the iteration with the ‘reconstructed’ OTU, Anakamacops petrolicus, K. acervalis, and Zygosaurus lucius form branches of a cacopine polytomy with Cacops.

Figure 7 Strict consensus trees for Analysis 1A (all scored olsoniforms) following removal of wildcard taxa from MPTs.

(A) Resultant tree from the iteration using the ‘conservative’ OTU of Kamacops acervalis; (B) resultant tree from the iteration using the ‘reconstructed’ OTU of K. acervalis. Only Dissorophidae is depicted in part B because the remainder of the consensus tree was unchanged from part A. Colors and symbols as with Fig. 6.

Analysis 1B (wildcard removal)

The iteration derived from the previous analysis with the ‘conservative’ OTU of Kamacops acervalis (which excluded this OTU) recovered 64 MPTs with a length of 329 steps (CI = 0.380; RI = 0.610; Fig. 8A). The omission of wildcards produced a weakly resolved topology that recovered all nominal dissorophids and all nominal trematopids in their respective clades but not Olsoniformes. Trematopidae is largely unresolved, with Acheloma cumminsi + Phonerpeton pricei as the only recovered relationship. Conjunctio multidens and all three species of Cacops are recovered as cacopines, with Cacops aspidephorus and Cacops morrisi as sister taxa, Cacops woehri as the sister taxon to this pair, and Co. multidens as the sister taxon to Cacops. All four species of Broiliellus, Diploseira angusta, and Dissorophus multicinctus are recovered as dissorophines in an unresolved polytomy. All amphibamiforms (including “Broiliellus” hektotopos) are single branches in a polytomy with Dissorophidae and Trematopidae. Bremer support is usually below three, and bootstrapping was usually below or just slightly above 50%.

Figure 8 Strict consensus trees for Analysis 1B (all scored olsoniforms) following removal of wildcard taxa from the matrix.

(A) Resultant tree from the iteration using the ‘conservative’ OTU of Kamacops acervalis; (B) resultant tree from the iteration using the ‘reconstructed’ OTU of K. acervalis. Only Dissorophidae is depicted in part B because the remainder of the consensus tree was unchanged from part A. Colors and symbols as with Fig. 6.

The iteration derived from the analysis with the ‘reconstructed’ OTU of Kamacops acervalis (which included this OTU) recovered 72 MPTs with a length of 338 steps (CI = 0.370; RI = 0.617; Fig. 8B). The strict consensus topology is nearly identical to that of the previous iteration with respect to mutually overlapping taxa. Anakamacops petrolicus, Kamacops acervalis, and Zygosaurus lucius do not form a clade (as they did in Liu, 2018) but instead form single branches of a polytomy with the clade of Cacops. Conjunctio multidens is still recovered as the earliest-diverging cacopine. Bremer and bootstrap support are consistently low. One node dropped below the threshold of strong support for Bremer decay index (Cacops aspidephorus + Cacops morrisi), and three nodes dropped below the threshold for bootstrapping (the two nodes of Cacops and Dissorophidae).

Analysis 2 (best representatives)

The analysis recovered 785 MPTs with a length of 382 steps (CI = 0.327; RI = 0.582; Fig. 9). The strict consensus is largely unresolved. Dissorophoidea is recovered (inclusive of Perryella olsoni), but the only in-group clades are a clade of the three species of Cacops + Anakamacops petrolicus (Cacopinae) and the pairing of Acheloma cumminsi + Phonerpeton pricei (Trematopidae). Both Dissorophidae and Dissorophinae therefore include only Dissorophus multicinctus. Bremer and bootstrap support are low except for Trematopidae.

Figure 9 Strict consensus tree for Analysis 2 (best representatives) from the iteration using the ‘conservative’ OTU of Kamacops acervalis.

The iteration with the ‘reconstructed’ OTU recovered the same topology with only minor deviations in bootstrap support and is not figured here. Colors and symbols as with Fig. 6.

Analysis 3

The iteration with the ‘conservative’ OTU of Kamacops acervalis recovered 420 MPTs with a length of 375 steps (CI = 0.333; RI = 0.589; Fig. 10). The strict consensus is largely unresolved and does not recover Amphibamiformes or Olsoniformes. Perryella olsoni is the sister taxon to Xerodromes. All nominal trematopids form a clade. Dissorophidae and Dissorophinae include only Dissorophus multicinctus, and Cacopinae includes only Cacops aspidephorus. All other dissorophids, amphibamiforms, and Palodromeus bairdi are single branches of a xerodrome polytomy with one branch for Trematopidae. Within Trematopidae, Mattauschia laticeps diverges at the base, followed by successively diverging branches of: (1) Anconastes vesperus + Tambachia trogallas; (2) Ecolsonia cutlerensis; and (3) Fedexia striegeli, the last of which is the sister taxon to Acheloma cumminsi + Phonerpeton pricei. Both Bremer and bootstrap support are universally low.

Figure 10 Strict consensus trees for Analysis 3 (dissorophid-focused).

(A) Resultant tree from the iteration using the ‘conservative’ OTU of Kamacops acervalis; (B) resultant tree from the iteration using the ‘reconstructed’ OTU of K. acervalis. Colors and symbols as with Fig. 6.

The iteration with the ‘reconstructed’ OTU of Kamacops acervalis recovered 23 MPTs with the same length of 375 steps. The strict consensus topology is more resolved with respect to nominal dissorophids, with all taxa except Reiszerpeton renascentis recovered as a clade. Platyhystrix rugosa is recovered as the earliest diverging dissorophid. All species of Broiliellus and Diploseira angusta are now recovered as dissorophines; Cacopinae remains restricted to Cacops aspidephorus. Bremer support did not change from the previous iteration, and bootstrap support changed only by 1–2% for nodes shared between iterations. Support for newly recovered nodes is extremely low, and most were not even recovered in the bootstrap tree (<1% occurrence).

Analysis 4

The iteration with the ‘conservative’ OTU of Kamacops acervalis recovered 220 MPTs with a length of 280 steps (CI = 0.436; RI = 0.649; Fig. 11). The strict consensus is poorly resolved. A noteworthy result is the recovery of the holotype of Conjunctio multidens (FMNH UC 673) at the base of Dissorophidae, whereas the two referred specimens (CM 91215, UCMP 40103) are recovered as sister taxa at the base of Cacopinae. The separation of the holotype from UCMP 40103 (historically the “Rio Arriba Taxon”) is the same as that of Schoch (2012). All other cacopines form a single polytomy with the exception of the pair of two specimens of Cacops woehri, the holotype (OMNH 73216) and a larger referred specimen (BMRP 2007.3.5). Bremer and bootstrap support are low for dissorophid nodes.

Figure 11 Strict consensus tree for Analysis 4 (specimen-level OTUs).

Node labels, colors, and symbols as with Figs. 6, 7. Asterisk (*) denotes a holotype.

The iteration with the ‘reconstructed’ OTU of Kamacops acervalis recovered 40 MPTs with a length of 281 steps (CI = 0.434; RI = 0.647). The strict consensus remains unchanged and is thus not depicted separately here. Bremer and bootstrap support are essentially unchanged.

Analysis 5 (taxon mirror of Dilkes (2020))

This analysis recovered 750 MPTs with a length of 348 steps (CI = 0.494; RI = 0.579; Fig. 12A). The strict consensus topology is discordant with that of Dilkes. Amphibamiformes and Olsoniformes are not recovered, and Dissorophidae and Dissorophinae only include Dissorophus multicinctus. Most nominal dissorophids are single branches in a polytomy with non-dissorophids. Cacopinae includes the three species of Cacops, Anakamacops petrolicus, Kamacops acervalis, and Zygosaurus lucius; within this, Cacops is monophyletic, and the other three taxa are single branches of a polytomy. All nominal trematopids form a clade, with one resolved in-group: Fedexia striegeli as the sister taxon to Acheloma cumminsi + Phonerpeton pricei. The other three taxa form a polytomy at the base. Bootstrap and Bremer support were usually below meaningful thresholds except for Trematopidae. The Adams consensus (not presented here) identifies Aspidosaurus binasser, Platyhystrix rugosa, Reiszerpeton renascentis, and Scapanops neglectus as wildcards following the same approach as in Analysis 1, but does not identify Brevidorsum profundum or Conjunctio multidens as wildcards (both were wildcards in Dilkes’ analysis).

Figure 12 Strict consensus trees for Analysis 5 (taxon mirror of Dilkes (2020)).

(A) Tree resulting from the analysis of this study’s matrix with the same taxon sample as Dilkes (2020); (B) tree resulting from the analysis of the same matrix without the four wildcard taxa identified by Dilkes (Brevidorsum, Conjunctio, Reiszerpeton, Scapanops). Colors and symbols as with Fig. 6.

Recently, Silva & Wilkinson (2021) proposed a method of representing consensus topologies when there are multiple islands recovered by a heuristic search by computing a consensus topology for each island. This can be useful when the strict consensus of all MPTs is relatively unresolved, although it relies on the analytical program not only identifying the number of islands but also the constituent MPTs. To the best of my knowledge, TNT does not do so; Serra Silva & Wilkinson used PAUP*, as in this analysis. The above analysis recovered three distinct islands, with 564 MPTs, 108 MPTs, and 78 MPTs (Fig. 13). When the strict consensus of each island is computed, the resolution is improved from the total consensus. The largest island (Fig. 13A) newly recovers Olsoniformes, a more inclusive Dissorophidae of all nominal taxa except Reiszerpeton renascentis, and a more inclusive Dissorophinae (Broiliellus, Diploseira angusta, Dissorophus multicinctus). Reiszerpeton renascentis is instead recovered as the sister taxon to Olsoniformes. For this island, only Brevidorsum profundum and Platyhystrix rugosa are identified as wildcards among Dissorophidae. The second largest island (Fig. 13B) newly recovers Trematopidae inclusive of R. renascentis and a relatively inclusive Dissorophidae (to the exclusion of R. renascentis, Bre. profundum, and Platyhystrix rugosa). Cacopinae now includes Conjunctio multidens, which is recovered at the base of the clade. Broiliellus brevis and Broiliellus texensis are recovered as sister taxa within Dissorophinae. For this island, no dissorophids (or olsoniforms) would be identified as wildcards. The smallest island (Fig. 13C) differs more starkly from the other two islands’ consensus topologies. Reiszerpeton renascentis forms a polytomy with Dissorophidae and Trematopidae, and C. multidens and P. rugosa are recovered as sister taxa within Dissorophinae. Trematopidae is fully resolved here, with Anconastes vesperus + Tambachia trogallas as the sister group to Ecolsonia cutlerensis. For this island, Aspidosaurus binasser, R. renascentis, and Scapanops neglectus would be identified as wildcards.

Figure 13 Strict consensus trees of individual islands recovered in Analysis 5.

(A) Island 1, consisting of 564 MPTs (trees 79–642); (B) Island 2, consisting of 108 MPTs (trees 643–750); (C) Island 3, consisting of 78 MPTs (trees 1–78). Colors and symbols as with Fig. 6.

The iteration with Dilkes’ wildcards removed recovered 296 MPTs with a length of 318 steps (CI = 0.522; RI = 0.597; Fig. 12B). All MPTs belong to the same island. The strict consensus topology is more congruent with that of Dilkes, although still with less resolution. Xerodromes, Olsoniformes, Trematopidae, and Dissorophidae are all recovered. The composition of Cacopinae and Dissorophinae is the same as that of Dilkes’ analysis. Platyhystrix rugosa and Aspidosaurus binasser are recovered as the earliest diverging dissorophids. Polytomies are found at the base of Trematopidae and Cacopinae, and Dissorophinae is a single polytomy. Nodal support remained low.

Analysis 6

The rerun of my original trematopid matrix with the newly combined Acheloma cumminsi OTU and the exclusion of Phonerpeton whitei recovered 27 MPTs with a length of 209 steps (CI = 0.445; RI = 0.615; Fig. 14A). For comparison, the original analysis recovered 105 MPTs with a length of 210 steps (CI = 0.443, RI = 0.640). The strict consensus topology is nearly unchanged; Olsoniformes and Amphibamiformes are recovered, but Xerodromes is not. Dissorophidae and Trematopidae include all of their respective nominal taxa. Within Trematopidae, the only resolved relationships are the pairing of A. cumminsi + Phonerpeton pricei and Rotaryus gothae as the sister group to this pair. The newfound resolution of R. gothae from the previously large polytomy is an intuitive result given the taxon sample modifications that were made to long-snouted taxa. Nodal support is weak except for some non-trematopid nodes.

Figure 14 Comparison of strict consensus topologies for Analysis 6 (taxon mirror of Gee (2020b)).

(A) Tree resulting from the analysis of Gee’s (2020b) trematopid-focused matrix with a combined OTU of the previous scorings of Acheloma cumminsi and Acheloma dunni and with the exclusion of Phonerpeton whitei; (B) tree resulting from the same taxon sampling and the updated matrix of this study. Colors and symbols as with Fig. 6.

The mirror analysis with the same taxon treatment and the revised matrix’s character sampling recovered 19 MPTs with a length of 282 steps (CI = 0.429; RI = 0.567; Fig. 14B). The strict consensus is slightly more resolved, but with the major caveat that Trematopidae only includes Acheloma cumminsi and Phonerpeton pricei. Anconastes vesperus + Tambachia trogallas is also recovered and forms one branch of a polytomy with the relatively exclusive Trematopidae, all other nominal trematopids, and Dissorophidae. Examination of the MPTs reveals that all nominal trematopids form a clade in all but one MPT in which Dissorophidae nests within Trematopidae as the sister group to Ecolsonia cutlerensis to form a late-diverging clade. Nodal support has improved, including a change in Bremer support for Olsoniformes and Trematopidae (from two to three) that reaches the threshold for meaningful support.

Analysis 7

Analysis of the updated Dilkes’ matrix recovered substantially more MPTs than the original analysis. With all 29 taxa, the analysis recovered 3,408 MPTs with a length of 169 steps (CI = 0.527; RI = 0.733; Fig. 15B), compared to 513 MPTs with a length of 163 steps (CI = 0.5460; RI = 0.7574; Fig. 15A). Recovering Dilkes’ original topology requires an additional 34 steps in the revised version of the matrix. All MPTs belong to the same island (both in Dilkes’ original analysis and in the new one). Resolution is relatively low, although not too dissimilar from Dilkes’ topology in this regard. Dissorophinae is restricted to Dissorophus multicinctus, and Cacopinae is restricted to the three species of Cacops. Cacops aspidephorus and Cacops morrisi are sister taxa to the exclusion of Cacops woehri, in contrast to the historic polytomy of these taxa, which form the entirety of Cacopinae here. All other dissorophids with the exception of Platyhystrix rugosa (the earliest diverging dissorophid) form a single polytomy. Bremer and bootstrap support have generally slightly declined for nodes shared between the original and the reanalysis, and Bremer support is below the threshold of meaningful support for almost all nodes. The resultant Adams consensus would identify a slightly different set of wildcards (Aspidosaurus binasser and Diploseira angusta and not Scapanops neglectus) than in Dilkes’ analysis, but I reran the search after removing the same four taxa that he identified as wildcards (Brevidorsum profundum, Conjunctio multidens, Reiszerpeton renascentis, Scapanops neglectus).

Figure 15 Comparison of strict consensus topologies for Analysis 7 (updated matrix from Dilkes (2020)).

(A) Tree resulting from Dilkes (2020) original analysis with the full taxon sample; (B) tree resulting from the analysis of the updated matrix with the full taxon sample; (C) tree resulting from Dilkes’ original analysis without the four wildcard taxa that he identified (Brevidorsum, Conjunctio, Reiszerpeton, Scapanops); (D) tree resulting from the reanalysis of the updated matrix without his wildcard taxa. Colors and symbols as with Fig. 6.

With removal of Dilkes’ wildcards, the analysis recovered 20 MPTs with a length of 151 steps (CI = 0.556; RI = 0.750; Fig. 15D), in contrast to the original 27 MPTs with a length of 143 steps (CI = 0.5874; RI = 0.7839; Fig. 15C). Dilkes’ original topology requires an additional 6 steps in the revised version of the matrix. The MPTs are evenly divided between two islands, but the strict consensus topologies of each are nearly identical, differing only in the relationships of dissorophines (Fig. 16). The strict consensus is more resolved than the previous iteration but less resolved than Dilkes’ topology. The relationships of Cacops are as with the previous iteration, but Anakamacops petrolicus, Kamacops acervalis, and Zygosaurus lucius are also recovered as cacopines. A more inclusive Dissorophinae is also recovered, with Broiliellus olsoni recovered as the earliest diverging taxon, followed by B. reiszi and then a polytomy of all other nominal dissorophines, which differs from the original analysis (Fig. 15C). This polytomy is the only source of differences between islands; in one, Dissorophus multicinctus and Diploseira angusta are sister taxa, and Broiliellus brevis and Broiliellus texensis are sister taxa (Fig. 16A). In the other island, Dis. multicinctus and B. texensis are sister taxa, and Dip. angusta and B. brevis are sister taxa (Fig. 16B). None of these configurations of Dissorophinae have been previously recovered. Acheloma cumminsi and Phonerpeton pricei are recovered in a polytomy with Anconastes vesperus + Tambachia trogallas. Fedexia striegeli and Ecolsonia cutlerensis are successive branches at the base of Trematopidae. Bremer and bootstrap support have again declined slightly for nodes shared with Dilkes’ analysis.

Figure 16 Strict consensus trees of individual islands recovered in Analysis 7.

(A) Island 1, consisting of 10 MPTs (trees 11–20); (B) Island 2, consisting of 10 MPTs (trees 1–10). Colors and symbols as with Fig. 6.

Analysis 8

The parallel analysis of the trematopid-focused sample in PAUP* recovered 23 MPTs with a length of 326 steps (CI = 0.506; RI = 0.567; Fig. 17A), compared to the 19 MPTs with a length of 282 steps (CI = 0.429; RI = 0.564; Fig. 17B) that I recovered in Analysis 6 (reproduced from Fig. 14B). The strict consensus topology is identical, but it is notable that there are three islands (of sizes 14, 8, and 1 MPTs), which differ in the relationships of olsoniforms (Fig. 18). As with the TNT results in Analysis 6, only one MPT does not recover all nominal trematopids within a clade, and that MPT instead recovers the nominal trematopids as a grade from Eoscopus lockardi to the nominal dissorophids (Fig. 18C). Based on the definition of Dissorophidae utilized here, several nominal trematopids are technically dissorophids (Actiobates peabodyi, Anconastes vesperus, Ecolsonia cutlerensis, Mordex calliprepes, Rotaryus gothae, and Tambachia trogallas). Similarly, the definition of Olsoniformes utilized here would result in a recovery of Mattauschia laticeps outside of Olsoniformes. The observation of multiple islands in the PAUP* analysis suggests that the MPTs of the TNT analysis might also be distributed across multiple islands. Nodal support is the same with respect to Bremer decay indices but is consistently higher in the PAUP* analysis, with differences ranging from 7% (Cacops morrisi + Cacops woehri) to 27% (Olsoniformes). Two nodes would be considered well-supported by bootstrap values in the PAUP* analysis but not in the TNT analysis: Dissorophoidea and Olsoniformes. I note that the PAUP* branch-and-bound search took a particularly long time (33.5 h) despite the eventual low number of MPTs and indicates the upper threshold of reasonable computation time for this matrix (23 taxa, 109 characters). Over half of the search process occurred after all MPTs had been recovered. A heuristic search with 10,000 random addition sequence replicates, holding 10 trees per step, recovered the same set of MPTs in just over a minute.

Figure 17 Comparison of strict consensus topologies for Analysis 8 (comparison of TNT and PAUP* on a trematopid-focused sample using this study’s revised matrix from Gee (2020b)).

(A) Tree resulting from the analysis using TNT (repeated from Fig. 14B); (B) tree resulting from the analysis with PAUP*. Colors and symbols as with Fig. 6.

Figure 18 Strict consensus trees of individual islands recovered in Analysis 8.

(A) Island 1, consisting of 14 MPTs (trees 1–14); (B) Island 2, consisting of eight MPTs (trees 15–22); (C) Island 3, consisting of one MPT (tree 23). Colors and symbols as with Fig. 6.

Analysis 9

The parallel analysis of Dilkes’ (2020) original matrix with all 29 taxa and in TNT recovered 23 MPTs with a length of 157 steps (CI = 0.529; RI = 0.757; Fig. 19B), in contrast to the original study using PAUP* (513 MPTs of length 163 steps; CI = 0.5460; RI = 0.7574; Fig. 19A). The strict consensus is identical to that of Dilkes’, as is the Bremer support, but bootstrap support is distinctly lower for all nodes, similar to the previous analysis. Here, the differences between the two analyses range from 3% (Anconastes vesperus + Tambachia trogallas) to 34% (Kamacops acervalis + Zygosaurus lucius). Five nodes dropped below the 50% threshold in the TNT analysis. The Adams consensus (not depicted) would identify the same four wildcard taxa as Dilkes’ original analysis (Brevidorsum profundum, Conjunctio multidens, Reiszerpeton renascentis, Scapanops neglectus).

Figure 19 Comparison of strict consensus topologies for Analysis 9A (comparison of TNT and PAUP* with the original matrix of Dilkes (2020)).

(A) Tree resulting from Dilkes (2020) original analysis in PAUP* with the full taxon sample; (B) tree resulting from the same matrix analyzed with TNT; (C) tree resulting from Dilkes’ original analysis without the four wildcard taxa that he identified (Brevidorsum, Conjunctio, Reiszerpeton, Scapanops); (D) tree resulting from the same matrix analyzed with TNT. Colors and symbols as with Fig. 6.

The TNT analysis of the matrix without Dilkes’ wildcards recovered a single MPT with a length of 142 steps (CI = 0.585; RI = 0.780; Fig. 19D), in contrast to Dilkes’ analysis in PAUP* (27 MPTs with a length of 143 steps; CI = 0.5874; RI = 0.7838; Fig. 19C). The differences relate only to resolution of polytomies recovered by Dilkes; there are no shifts in the general position of taxa. The same pattern of Bremer and bootstrap support was identified, with the latter ranging between 4% (Anconastes vesperus + Tambachia trogallas) and 20% (all three species of Cacops; Dissorophinae; Cacopinae + Dissorophinae). However, zero nodes dropped below the 50% threshold in the TNT analysis.

The TNT analysis of my updated version of Dilkes’ matrix with all taxa sampled recovered 284 MPTs with a length of 161 steps (CI = 0.509; RI = 0.734; Fig. 20A). Resolution has decreased across the tree with a large basal polytomy in Dissorophidae. Cacopinae consists only of the three species of Cacops. Platyhystrix rugosa is still recovered as the earliest-diverging dissorophid. Trematopidae has also lost resolution; only Anconastes vesperus + Tambachia trogallas is recovered. Bremer and bootstrap values tend to be lower for overlapping nodes. The Adams consensus (not depicted) does not identify the same four wildcards as Dilkes (2020). Of those four, only Reiszerpeton renascentis is still recovered in a wildcard position, in addition to Aspidosaurus binasser and Diploseira angusta, which were not wildcards in Dilkes’ original analysis.

Figure 20 Comparison of strict consensus topologies for Analysis 9B (comparison of the original and updated versions of the matrix of Dilkes (2020), using TNT).

(A) Tree resulting from the reanalysis of the updated version of Dilkes (2020) matrix with the full taxon sample; (B) tree resulting from the analysis of the original version of Dilkes’ matrix (repeated from Fig. 19B); (C) tree resulting from the reanalysis of the updated version of Dilkes (2020) matrix without the four wildcard taxa that were identified by Dilkes (Brevidorsum, Conjunctio, Reiszerpeton, Scapanops); (D) tree resulting from the analysis of the original version of Dilkes’ matrix (repeated from Fig. 19D). Colors and symbols as with Fig. 6.

The TNT analysis of the same matrix without Dilkes’ wildcards recovered 12 MPTs with a length of 148 steps (CI = 0.554; RI = 0.778; Fig. 20C). The tree is more resolved than in the previous iteration, with two cacopine clades: (1) all three species of Cacops; and (2) Anakamacops petrolicus + Kamacops acervalis + Zygosaurus lucius. This is the only analysis to recover all three taxa within Kamacopini (the clade defined as all taxa closer to K. acervalis than to Cacops aspidephorus; Liu, 2018). The arrangement of Dissorophinae, with Broiliellus olsoni as the earliest diverging taxon, followed by Broiliellus reiszi, differs from both the previous iteration with all taxa sampled and from previous studies. Within Trematopidae, Fedexia striegeli and Ecolsonia cutlerensis are now recovered as successively diverging taxa at the base of the clade rather than as exclusive sister taxa. Acheloma cumminsi and Phonerpeton pricei are also not recovered as exclusive sister taxa and instead form a polytomy with the pair of Anconastes vesperus + Tambachia trogallas. Bremer and bootstrap values are again lower compared to the original matrix analyzed in TNT.

Discussion

The original goal of this study was to expand my previous trematopid matrix to encompass dissorophids as a means of independently testing the relationships of dissorophids and olsoniforms more broadly. This study “accomplished” that goal, but as may be apparent from a first-hand examination of the results and as I hope to make a case for, the results of both this study and those of previous studies should be treated cautiously. This discussion is therefore divided into three main sections: (1) a brief discussion of the first-hand results of the analyses performed here; (2) a broader discussion of topological disparity, identifiable sources of this disparity, and a critical examination of previous methodologies; and (3) a summary of the state of affairs in dissorophoid phylogenetics and taxonomy with some highlighted areas for future study.

Tallying topologies

This section summarizes the key findings from the various analyses that were performed in this study as they relate to the recovered topologies. It comes as little surprise that across my nine analyses, there is little consensus; indeed, some analyses recover little resolution at all. Larger taxon samples produced less resolution and rarely recovered all nominal dissorophids or all nominal trematopids in a clade (e.g., Analyses 1A, 2; Figs. 6, 9). Parioxys bolli is unsurprisingly not recovered as a dissorophid, let alone as a dissorophoid (Fig. 6). The skew towards cranial characters in this matrix, a characteristic of other matrices as well, reflects the precedent of using cranial remains to differentiate taxa. Even with the expanded postcranial character sampling of this study’s matrix, most characters cannot be scored for this taxon. “Broiliellus” hektotopos is also not recovered as an olsoniform, let alone as a dissorophid (Figs. 6–9). Amphibamiformes is never recovered when this taxon is sampled, and it always forms a polytomy with other amphibamiforms. The lack of a monophyletic Amphibamiformes in most analyses likely reflects the character sampling being derived from olsoniform matrices and thus undersampling characters that capture amphibamiform synapomorphies.

A more surprising result is the position of Reiszerpeton renascentis as an unplaced olsoniform or outside of a clade of all other dissorophids in some analyses (Analyses 3, 5; Figs. 6, 10B, 13), as it was always recovered as a dissorophid in previous studies (Figs. 3, 4). One possibility is that this small specimen was very immature and therefore shares certain qualitative features with amphibamiforms; its original identification as a specimen of Tersomius texensis speaks to this (Maddin et al., 2013). With that said, it never clusters with amphibamiforms. There may be enough features shared with most/all dissorophids or olsoniforms (e.g., a septomaxilla at the mid-length of the naris; postorbital lacking a markedly offset posterior terminus) to maintain its olsoniform affinities. The apparent absence of a ventral process of the prefrontal (Maddin et al., 2013:454) may also contribute to its stemward slippage; this process (the VPP) is found in all other dissorophoids sampled here.

A different hypothesis is that full sampling of trematopids, including the rarely sampled Carboniferous taxa like Mordex calliprepes, draws the taxon down. Many of these Carboniferous taxa have a relatively higher number of plesiomorphies than the more commonly sampled taxa. This hypothesis is supported by some MPTs of Analysis 5 in which Reiszerpeton renascentis is recovered as an early diverging trematopid (Fig. 13B). However, it never clusters with trematopids in analyses of the original or revised matrix of Dilkes (2020), which use the same taxon sample (Analyses 7, 9; Figs. 15, 16, 19, 20). These discrepancies indicate that character sampling and construction are more likely an explanator since scoring approaches between my matrix and my revised version of Dilkes’ were the same (e.g., the taxon cannot be scored for characters invoking skull length). The most likely scenario in my opinion is that the missing data for R. renascentis are responsible for its peculiar position. The holotype is only a partial skull, incomplete posteriorly, and definitively lacks apomorphic ornamentation found in cacopines and dissorophines. Therefore, its combination of known scores may approximate the “ancestral” olsoniform condition, even though there is no evidence of either trematopid apomorphies or conditions contrary to the diagnosis of Dissorophidae.

This conjecture could also apply to Brevidorsum profundum and Platyhystrix rugosa. Brevidorsum profundum is represented only by a fragmentary partial skull (without apomorphic ornamentation) and a few postcranial fragments (without osteoderms). Hook (1989) even suggested that it might be a small trematopid related to what was eventually placed in Phonerpeton. However, B. profundum is either recovered with other dissorophids to the exclusion of Reiszerpeton renascentis (e.g., Analysis 3; Fig. 10B) or in a polytomy of all dissorophids including R. renascentis and perhaps other taxa (e.g., Fig. 10A). A few features in which B. profundum is similar to most other olsoniforms and differs from R. renascentis (e.g., tabular-squamosal contact; semilunar curvature) are evidently sufficient to avoid the same degree of stemward slippage.

Platyhystrix rugosa is represented only by a badly crushed skull and isolated postcrania (Berman, Reisz & Fracasso, 1981). It was identified as a wildcard taxon in Analysis 1A (Fig. 6) and was not always recovered as a dissorophid (e.g., Analyses 3, 7, 9, some MPTs of Analysis 5; Fig. 13B), like Brevidorsum profundum. It too never exhibits the same slippage as Reiszerpeton renascentis and is often recovered as the earliest-diverging dissorophid (e.g., Analysis 5; Figs. 10B, 12B, 13B, 15, 19, 20). One consideration worth future consideration is whether this position is still artificial, even if its position within Dissorophidae is secure. The only semi-complete skull of P. rugosa is badly crushed. While mostly complete longitudinally, the naris is not sufficiently preserved to be certain of its shape or the relationship of the elements that normally frame it (e.g., lacrimal; Berman, Reisz & Fracasso, 1981). Additionally, P. rugosa has historically been scored as lacking osteoderms, contrary to all other nominal dissorophids; the condition of its spines is therefore regarded as ornamentation of the spines, rather than a separate dermal ossification. Notably, however, Witzmann & Soler-Gijón (2010), the only published histological study of these spines, termed them as osteoderms. Dilkes’ (2020) analysis of his matrix with and without osteoderm characters did result in a slight change in topology, so the identity of the spinal ornamentation has clear phylogenetic implications as well. Without these characters, P. rugosa formed a polytomy with Aspidosaurus binasser and a branch for all other dissorophids, rather than diverging before A. binasser (Dilkes, 2020). It is possible that the combination of absent osteoderms and missing data may produce an artificially early-diverging position, as the taxon is difficult to differentiate from trematopids in scoring (e.g., the nares cannot be characterized) and it lacks certain dissorophid apomorphies (e.g., osteoderms). The Early Permian age of the material from which most scores are derived is incongruent with its present early-diverging position.

There is usually poor resolution within Trematopidae except in analyses with restricted taxon samples (Fig. 5). In this study, nominal trematopids are recovered as a clade in most analyses except Analyses 1A and 2, which recovered almost no resolution anywhere (Fig. 9). A restricted subset of seven trematopids in Analysis 3 did form a clade in spite of relatively little resolution elsewhere (Fig. 10). The more interesting result is that a trematopid-focused sample did not recover all nominal trematopids in a clade (Analyses 6 and 8; Figs. 14, 17). This pattern persisted even when examining individual tree islands of Analysis 8 (Fig. 18), although as noted in the Results, 18 of the 19 MPTs of Analysis 6 did recover all nominal trematopids in a clade. One explanation may be that two characters were removed to avoid redundancy/parsimony-uninformative characters (narial elongation, lacrimal-naris; Appendix 2). Trematopids were all scored the same for both characters, and this redundancy may have overweighted certain apomorphies (the lacrimal always enters an elongate naris in this clade). Another is that the net addition of characters (+24 compared to Gee (2020b)), many of which are postcranial characters for which most olsoniforms cannot be scored, creates sufficient uncertainty to preclude recovery of all nominal taxa in a clade. The commonly recovered in-group nodes are Acheloma cumminsi + Phonerpeton pricei, a longstanding relationship between two Permian taxa from Texas (Figs. 7–12, 14, 17, 19), and Anconastes vesperus + Tambachia trogallas, a more perplexing relationship (Late Carboniferous of New Mexico and Early Permian of Germany; Figs. 10, 12B, 14B, 15, 17, 19, 20) but one that has been recovered in most previous studies (Fig. 2).

The resolution of Dissorophidae is quite variable, especially with respect to historical wildcard taxa or historically unsampled taxa. All nominal dissorophids form a clade only in restricted taxon samples that omit wildcards like Reiszerpeton renascentis (e.g., Analyses 1B, 2, 5; Figs. 7, 8, 12B) or that have a more limited outgroup sample (e.g., Analyses 7, 9; Figs. 15, 19, 20). Where resolution is appreciable, Platyhystrix rugosa and Aspidosaurus binasser usually form successively diverging branches at the base as they do in practically all other dissorophid analyses (Figs. 3, 4; see Figs. 12B, 15D, 19B, 19D, 20C, for topologies of this study).

Dissorophinae is an “all or nothing” of sorts; either it is restricted to Dissorophus multicinctus (Figs. 9, 10A, 12A, 15B, 19B, 20A) or it includes this taxon, all valid species of Broiliellus, and Diploseira angusta (Figs. 7, 8, 10B, 12B, 15D, 19D, 20B). The more inclusive composition follows the historical characterization of Dissorophinae. This includes Broiliellus arroyoensis, which has never been previously sampled because the cranial sutures are entirely unknown. It is united with other nominal dissorophines here by its apomorphic ornamentation and demonstrates that taxa without sutures or with a high proportion of missing data are not assured to be wildcards (e.g., Kearney, 2002; Kearney & Clark, 2003; Wiens, 2003a, 2003b, 2005, 2006; Wilkinson, 2003; Prevosti & Chemisquy, 2010; Wiens & Morrill, 2011; Guillerme & Cooper, 2016). Taxon removal exerts the strongest apparent influence on Dissorophinae; analyses with progressive taxon removal often produced a more inclusive clade (e.g., Analyses 5, 7; Fig. 12, 15). The interrelationships of dissorophines remain poorly resolved however.

Cacopinae is similar to Dissorophinae in either being monotaxic (Cacops aspidephorus) or in comprising a consistent, more exclusive clade. The more exclusive version of Cacopinae almost always includes all three species of Cacops (Figs. 15B, 20A); only in Analysis 3 is C. aspidephorus recovered as the sole cacopine (Fig. 10). Some analyses also recovered Anakamacops petrolicus, Kamacops acervalis, and Zygosaurus lucius as cacopines (Figs. 7B, 8B, 12, 15D, 19D, 20C). Conjunctio multidens was also sometimes recovered as a cacopine (Figs. 7, 8, 13B), a result sometimes previously found (Figs. 3, 4). Cacops was usually monophyletic (but see Analyses 1A and 2; Figs. 7B, 9), with Ca. aspidephorus and Cacops morrisi as sister taxa and Cacops woehri as the sister taxon to this pair.

Liu (2018) recovered the Middle Permian dissorophids Anakamacops petrolicus, Kamacops acervalis, and Zygosaurus lucius as a clade, which he termed Kamacopini and which was defined as the most inclusive clade that includes K. acervalis but not Cacops woehri. Subsequent studies have either recovered Kamacopini to the exclusion of A. petrolicus (Dilkes, 2020) or as a monotaxic clade with the nominal taxa as part of a larger polytomy (Gee et al., 2021; Figs. 7, 8, 10, 12, 15). Analysis 9A, with Dilkes’ wildcards excluded, is the only one to recover the same Kamacopini as Liu (Fig. 19). This likely results from the very poor characterization of Z. lucius, which does not differ from K. acervalis in scoring except for the distribution of missing data. Anakamacops petrolicus conversely differs from Z. lucius for one character and from K. acervalis by two or three depending on which OTU of the K. acervalis is used. Iratusaurus vorax and Nooxobeia gracilis, the other Middle Permian dissorophids, are too fragmentary and lacking in apomorphies to assess whether they are kamacopins (Gubin, 1980; Gee, Scott & Reisz, 2018; Fig. 6).

Collectively, the results reflect a few influential factors. The first is taxon sampling; it is not surprising that including many poorly known taxa (Analyses 1A, 2, 3) led to very poor resolution (Figs. 6, 9, 10). This pattern was also apparent in my trematopid study in which restricting the taxon sample to mirror that of previous studies led to the recovery of substantially more resolution than when all taxa were sampled (Gee, 2020b). Nonetheless, taxa like Broiliellus arroyoensis demonstrate that taxon sampling criteria must be nuanced and not rely on blanket characterizations such as “no cranial sutures known,” as a handful of qualitative features may still be highly informative. However, the presence of tubercular ornamentation in Iratusaurus vorax and Nooxobeia gracilis, a cacopine apomorphy, was insufficient to draw them into Cacopinae in any analysis. The absence of this ornamentation in Cacops woehri, which still clusters with the other species of Cacops, could be a confound. However, I. vorax and N. gracilis are also some of the most fragmentary taxa (scored for <20% of characters; Table 1). The status of osteoderms also predictably exerts influence (as shown also by Dilkes, 2020); as discussed above, nominal dissorophids that are sometimes recovered outside of Dissorophidae or at the base are often those for which the postcranial skeleton is not sufficient to score the presence/absence (e.g., Reiszerpeton renascentis) or where osteoderms are scored as being absent (e.g., Platyhystrix rugosa).

To summate, in spite of a few broadly conserved aspects (e.g., composition of Dissorophinae), there remain many outstanding questions and uncertainties, even if the discussion is restricted to relatively resolved, restricted-taxon-sample analyses (Figs. 12–20). Therefore, there is very little consensus in dissorophid intrarelationships, and as Dilkes (2020) remarked, it is not merely surprising but rather concerning that there is so much disparity between previous studies that used a nearly identical character matrix. This questions whether any study, including this one, has produced a reasonably robust topology that is acceptable for use in other studies or for qualitative discussion of olsoniform evolution. The following sections present a more detailed discussion of my findings regarding potential methodological explanators for the persistent disparity, some of which are rather concerning and some of which may invalidate previous analyses.

The search for a consensus

Almost every living worker who has specialized in terrestrial dissorophoids in the last four decades (J. Anderson; D. Berman; D. Dilkes; N. Fröbisch; B. Gee; R. Holmes; A. Huttenlocker; H. Maddin; A.R. Milner; J. Pardo; R. Reisz; R. Schoch) has participated in dissorophid phylogenetic analysis, with only two (Fröbisch, Schoch) involved in more than one analysis. However, almost every previous matrix other than the modified one used in this study is a direct derivate from Schoch (2012), and almost every one of these derivates is over 95% similar in character sampling, taxon sampling, and scoring to the original (Figs. 3, 4). Two of the three first-order derivates only added or subtracted taxa (Maddin et al., 2013; Schoch & Sues, 2013). The majority of cumulative changes present up through the derivate by Dilkes is the result of Dilkes adding seven new characters (cumulative changes summarized in Appendix 8). Gee et al.’s (2021) matrix is the most recent derivate to be published, although it is not a direct derivate, as it combined Holmes, Berman & Anderson’s (2013) derivate with the amphibamiform-focused matrix of Maddin et al. (2013); accounting for elimination of redundancies, only 33 characters are carried over from the former. Fröbisch & Reisz (2012) used a slightly modified version of Polley & Reisz’s (2011) trematopid matrix, although it only sampled five dissorophids. However, because all 53 characters of Polley & Reisz were incorporated into Schoch (2012), Fröbisch & Reisz’s matrix is in fact very similar in character sampling, even if it is not derived from Schoch’s. Schoch did not expressly state whether he rescored characters but the absence of any polymorphisms in his matrix in comparison to nine in Polley & Reisz’s matrix suggests as much. The Gee (2020b) trematopid matrix and the derivate used here are also in part derived from Polley & Reisz, so the general character sampling thus converges on the sampling of Schoch (2012). However, character construction differs for many (Appendix 1), and scoring was expressly novel for all characters. There are also notable departures from the sampling of previous analyses to avoid redundant or dependent characters (Appendices 1, 2).

In detailing this history, I want to emphasize that I am not advocating against the standard practice of propagating an existing matrix; there are many advantages to doing so. However, implicit in the propagation of a consensus matrix is the assumption that the underlying framework is largely sound. While there may be minor differences in scoring philosophy, such as whether reconstructions can be used, the matrix should obviously be as error-free as possible while maximizing the available data. If it is not, then widespread adoption of this matrix will result in widespread propagation of errors. There are therefore also inherent disadvantages to propagating matrices when their quality is suspect. This underscores the emphasis on quality of the matrix, rather than quantity of characters, duration or frequency of usage, or other quantitative metrics, that have been raised by numerous recent studies (e.g., Brazeau, 2011; Simões et al., 2017; Laurin & Piñeiro, 2018). However, wide discrepancies in practice clearly persist within the field. As reflected in Analysis 7 (Fig. 15, Appendices 5, 6), I have concerns about the accuracy of the widely propagated Schoch matrix. These findings therefore cast doubt on previous studies that used this matrix, which I outline in greater detail below.

Scoring issues

Examination of previous matrices for possible explanators of topological disparity led me to conduct a thorough survey of Dilkes’ version of the Schoch (2012) matrix (as the most recent direct derivate). I identified a large number of scores for features that are simply unknown in the given taxon (73 in total; Appendix 5). I also identified a smaller number of unscored cells that can definitively be scored from the literature (35 total) and scores that were scored for the wrong character state (35 total). I want to emphasize that the changes that I made are corrections to unequivocal errors in the sense that standard practices do not permit scoring of features that are unknown. I also accounted for the datedness of some literature with respect to previously unscored cells that were newly scored. While some typographical errors are to be expected in any matrix, the number of changed scores (143) and the pattern of these scores either suggest an intentionally non-standard coding philosophy that permits scoring of unknown features or an unusually large number of typographical errors. Most of these errors stem from either the original matrix or the first direct derivative (Holmes, Berman & Anderson, 2013), so they have been propagated through several analyses. Below I outline a few examples in detail.

Cacops provides the clearest example. For Cacops woehri, I identified 14 errors; in a matrix of 70–77 characters, this is a substantial amount (note that the particular scores of this taxon that have been propagated were introduced by Holmes, Berman & Anderson (2013)). A total of 10 of these relate to features that are simply not preserved (exoccipital, stapes, quadrate, postcrania). Having worked extensively on Richards Spur, from which C. woehri is known, I am not aware of any unpublished material of C. woehri that can reconcile these errors (but see Gee, Bevitt & Reisz, 2019:fig. 9.7–9.14 for an isolated jaw articulation that was assigned to Dissorophidae cf. C. woehri). At least the postcranial scores could not have been taken from the original scoring by Fröbisch & Reisz (2012) because that matrix has zero postcranial characters. One of the 14 errors is a score for the wrong state (tubercular ornamentation present); C. woehri lacks the tubercular ornamentation found in other species of Cacops (Fröbisch & Reisz, 2012; Fröbisch, Brar & Reisz, 2015; Gee, Bevitt & Reisz, 2019). Notably, of the 14 erroneous scores, 13 were scored identically to both of the other species of Cacops; the last score was only the same as Cacops morrisi (Cacops aspidephorus was unscored). In fact, there were no characters for which the three species differed other than in the distribution of missing data. This false homogeneity certainly accounts for the unresolved relationships of the three species in all previous analyses (Figs. 3, 4), despite the many features that separate them taxonomically (Fröbisch & Reisz, 2012; Gee & Reisz, 2018a) and the recent questioning of whether C. woehri is even properly placed in the genus (Anderson, Scott & Reisz, 2020). The most reasonable conclusion is that scores for C. woehri were “assumed” on the basis of the phenetic placement of the taxon within Cacops. This approach is problematic since a phylogenetic analysis should inform taxonomy, rather than vice versa. The notable cranial differences of C. woehri from the other two species suggest that it may have also differed in other skeletal attributes that are not presently known for it.

Cacops aspidephorus also corroborates the hypothesis that scores have been “assumed.” The taxon is famously known for the total absence of any knowledge of the cranial sutures (inclusive of the palate and braincase) due to poor preservation. Only in the past year has this gap been addressed (Anderson, Scott & Reisz, 2020). However, in Schoch’s (2012) original matrix, there were no fewer than 15 scores for which the sutures would have to be known in order to score the taxon. These scores were then propagated without modification by all derivates. Schoch did not personally examine the material of this taxon (as indicated in his Table 1), and it is telling that while he reconstructed the cranial anatomy of almost every dissorophid with their respective known sutures, that of C. aspidephorus is only a silhouette (see Figs. 6 and 7 therein). While Anderson, Scott & Reisz (2020) description corroborated many of these scorings, at least five original scores were shown to be errors (e.g., parasphenoid dentition, exoccipital-postparietal contact). One originally unscored character (postorbital-supratemporal) should have been scoreable if other cranial suture characters could have been scored. These erroneous scores are identical to those of C. morrisi and would have to have been “assumed” from this taxon (Cacops woehri was not sampled in the original matrix). This is further corroborated by the observation that Cacops morrisi does not show a pattern of erroneous scores, probably because it was the exemplar from which scores for the two other species were “assumed”; only one score is clearly erroneous: the iliac blade. No pelvis is known for the taxon, but this could have been “assumed” either based on the fact that this feature is an olsoniform apomorphy or based on Williston’s description of C. aspidephorus.

As discussed in the Methods and Appendix 3, the previous scoring for Kamacops acervalis cannot be fully corroborated by the literature. This taxon’s cranial sutures have never been described or figured, but it is scored for many characters that require these sutures to be known and is specifically scored almost identically to the three species of Cacops. The only reconstruction of the skull roof of Kamacops acervalis in dorsal view (Schoch, 2012:fig. 6) lacks sutures, like Cacops aspidephorus. If sutural characters could be scored for these taxa, it is unclear why they were not reconstructed with them. The scoring of my matrix is based on my assumption that the sutures are not actually known since data to support their characterization have never been published. Broiliellus, with four commonly sampled species, does not clearly show evidence of widespread “assumed” scores. Broiliellus arroyoensis, for which sutures are also unknown, has never been previously sampled or reconstructed, and it is almost never discussed in the literature. Broiliellus olsoni does have a few erroneous scores for elements that are not preserved, but no pattern is apparent for other species. Broiliellus brevis and Broiliellus texensis are difficult to assess from the literature given its datedness (Williston, 1914; Carroll, 1964a; DeMar, 1966b). These descriptions are not well-suited for scoring due to their brevity and limited figures. Schoch indicated that he personally examined material of both taxa, and therefore, it is possible that he was able to score features from personal examination that are not mentioned in the literature, contrary to C. aspidephorus.

Some of the errors that I identified are related to how characters are defined, something that Dilkes (2020:20–22) discussed at length. For example, the position of the jaw articulation is referenced by the position relative to the exoccipital facets. However, the character has been scored for taxa in which the jaw articulation or the exoccipitals are unknown (e.g., Cacops woehri, Scapanops neglectus, Tambachia trogallas). It was probably assumed that the occiput was vertical (like in most temnospondyls), from which it can be assumed that the occipital margin of the postparietals is an acceptable proxy for the exoccipitals’ posterior extent. If this series of working assumption was indeed used, the character should have been redefined (as I did here; Appendix 1). Otherwise, it could lead to inconsistent scoring where one worker operates with this unstated assumption and thus scores the character, but another worker operates strictly based on the character as defined and thus leaves it unscored. The other set of characters for which this applies are those related to the relative length or height of different regions of the skull (e.g., suborbital bar height, distance between the squamosal embayment and orbit). These are all defined by relation to the midline skull length, yet a number of taxa whose skulls are incomplete posteriorly were scored for these characters (e.g., Broiliellus olsoni, Reiszerpeton renascentis). These characters could be redefined based on a proxy to facilitate scoring of incomplete specimens (something that Liu (2018), attempted, but that Dilkes (2020), reversed). As defined, these characters cannot be scored from such specimens.

Certain characters also seem more susceptible to either “assumed” scoring, or in some instances, a peculiar lack of scoring. The best example of the latter is how exoccipital-tabular contact (only found in Sclerocephalus in the matrix) is often scored, but exoccipital-postparietal contact is not. This is in spite of frequent explicit description and figuring of the exoccipital-postparietal contact and the lack of an exoccipital-tabular contact. Some scores here also appear to have been “assumed” based on a taxon’s broader affinity (e.g., to Dissorophidae). For example, a prefrontal-postfrontal contact is not found in any dissorophid and only in the earliest-appearing trematopids (not sampled in Schoch’s matrix or any derivate). Anconastes vesperus is scored as lacking this contact, yet no specimen preserves the medial margins of the orbit (Berman, Reisz & Eberth, 1987). The dorsal quadrate process (a xerodrome feature as scored) is another example; several taxa without complete (or any) quadrates are scored as having this feature (e.g., Cacops woehri).

It is worth pointing out that many of the original erroneous scores, whether a typographic error, an assumption, or an inference, were not necessarily inaccurate. It is reasonable to predict, for example, that Cacops morrisi would have an expanded iliac blade, an olsoniform synapomorphy that is positively identified in Cacops aspidephorus (Williston, 1910). However, this feature has yet to be positively identified in Cacops morrisi and should not be scored as such. It is simply unknown. The correction of five previously inferred scores of C. aspidephorus following the study of Anderson, Scott & Reisz (2020) underscores the point that even scores that are inferred through close phenetic relatedness may prove to be wrong when data are produced to assess them. A phylogenetic analysis is a test of hypotheses that goes beyond the historical phenetic frameworks that were not only subject to, but predicated on, each worker’s personal conceptions of relatedness. While a phylogenetic analysis may be a test of inferences, it is not a test that can be based on inferences–it is a test only properly based on strict observations (i.e., data). I am under no illusions that phylogenetic analyses are unbiased or completely objective–taxon sampling and character construction are just two ways in which workers’ biases can be imposed upon the analysis. But there is an implicit aim for minimizing biases when conducting these analyses, and scores that are simply inferred based on phenetic taxonomic placement or previous phylogenetic placement are the opposite.

Correcting for these inferences or assumptions, as I have done, has predictably reduced resolution and nodal support (Figs. 15, 20). While loss of resolution and nodal support is an unfavorable outcome, an analysis based only on what is properly observable is the only truly defensible topology. Relationships recovered from inferred “observations” are spurious. A lack of resolution does not necessarily indicate that the matrix is compromised or poorly constructed, and workers should not tinker with the matrix in a way that produces resolution at the cost of data integrity. The ultimate goal is to recover correct clades while avoiding incorrect ones (Rineau, Zaragüeta i Bagils & Laurin, 2018), not to maximize resolution at the expense of accuracy. The sampled characters could capture all of the historical characters that were incorporated into phenetic taxonomy yet be insufficient to recover statistical support for relationships. Such a possibility is inherently tied to the data available for each taxon and thereby to the incomplete fossil record.

Slippery slope

The concerns I outlined in the matrix of Schoch (2012) and its derivates are not exclusive to this “family” of matrices. The 70 characters from Schoch’s (2012) matrix were carried over to Schoch’s (2018a) matrix, which broadly samples dissorophoids (Fig. 2). Presumably, the scores for the 10 taxa that are sampled in both matrices were also propagated; this appears to be the case based on a random assessment of cells that I corrected (i.e., the same erroneous scores are found in both matrices). The taxon sample of Schoch’s (2018a) matrix was skewed towards amphibamiforms (only eight olsoniforms are sampled), but this matrix has been expanded in four studies (Atkins, Reisz & Maddin, 2019; Gee & Reisz, 2019; Schoch, Henrici & Hook, 2020; Schoch & Milner, 2021), three of which added olsoniforms. The same matrix with a different taxon sample was also used by Schoch & Witzmann (2018) in their study of the micromelerpetid Limnogyrinus; this study preceded the online publication of Schoch (2018a) by a few weeks.

The densest taxon sampling of any previous study with respect to olsoniforms is that of Atkins, Reisz & Maddin (17 OTUs), but as far as I can discern, the scores for the first 70 characters for almost all newly added olsoniforms (Anconastes, Aspidosaurus, Brevidorsum, Kamacops, Platyhystrix, and the “Rio Arriba” and “Admiral” taxa) are taken directly from Schoch (2012). This would explain why the referred specimen of Conjunctio multidens is still treated as the “Rio Arriba Taxon,” separate from the holotype, and why Scapanops neglectus is still termed the “Admiral Taxon”; both terms went into disuse following Schoch & Sues (2013). Therefore, the same erroneous scores present in Schoch (2012) would also be present in Atkins, Reisz & Maddin’s matrix, which I cursorily confirmed based on a random examination of equivalent cells that I had corrected in Dilkes’ (2020) matrix. Workers seeking to expand any version of Schoch’s (2018a) matrix should carefully examine the matrix to assess the fidelity of scores in order to avoid the same issue of propagation of dubious scores. I want to emphasize that my discussion of previous matrices’ scores is not meant as overt criticism of other workers for the sake of being critical (it was not even part of my original study design), nor is it under any assumption that I or any other worker produce entirely infallible work. It is instead meant to highlight that historic practices have produced a topology that is not reproducible under best practices (e.g., scoring only based on observed features), with the topology resulting from correction of errors being noticeably different (Figs. 15, 20).

The treatment of polymorphisms

The treatment of polymorphisms has been a contentious topic in phylogenetics for some time (e.g., Wiens, 1999; Watanabe, 2015). Polymorphisms are generally rare in temnospondyl matrices. It was not until Liu (2018) that polymorphisms were introduced into a dissorophid matrix, and it was only for Conjunctio multidens. Dilkes (2020) subsequently introduced one polymorphic scoring for Dissorophus multicinctus. While some of this owes to the singleton representation of many taxa, there are also examples of incontrovertible biological variation within a single individual (e.g., postorbital-supratemporal contact in Phonerpeton pricei; Dilkes, 1990) that were not scored. This specific example is odd because Schoch (2012) explicitly called out this polymorphism (p. 128 therein) yet did not score it as such. A second example is his mention of the polymorphic state of the intertemporal in Sclerocephalus, another sampled taxon for which this polymorphism is unscored (p. 128 as well). In contrast to this propagated matrix, polymorphisms are widespread in my matrix; at the species level, there are 48 polymorphisms. Such disparity is not related to my increased sampling of postcranial characters; only two polymorphisms are scored for such characters.

It has historically been assumed that polymorphic characters are less reliable for inference (e.g., Wiens, 1995). Indeed, comparisons of matrices with polymorphisms with the same matrices without polymorphisms recover distinctly different topologies (e.g., Trinajstic & Dennis-Bryan, 2009; Watanabe, 2015; Garbin, Ascarrunz & Joyce, 2018). Nonetheless, not representing intraspecific variation in some form is an oversimplification of the data, and proper representation is essential, even if it comes at the expense of topological resolution. Furthermore, analysis of both simulated and empirical datasets has demonstrated that failing to score polymorphisms may in fact decrease accuracy (e.g., Wiens & Servedio, 1997, 1998; Wiens, 1998; Trinajstic & Dennis-Bryan, 2009). Therefore, there is no strong a priori standing for intentionally omitting polymorphisms. Numerous strategies for approaching polymorphisms have been discussed (e.g., Kornet & Turner, 1999; Wiens, 1999), but many are not well-suited for paleontological datasets. For example, scoring based on the frequency with which a certain state appears will require an appreciable sample size that is rarely met for extinct tetrapods. I believe that no olsoniform is known from such a sample size. The use of an alternative character state for polymorphisms (e.g., ‘scaled,’ ‘unscaled,’ and ‘unordered’ scoring; Campbell & Frost, 1993; Mabee & Humphries, 1993; Wiens, 1995, 1999) is one option that does not rely on a large sample. However, no previous olsoniform study has utilized this approach (the use of ‘a’ for polymorphisms by Polley & Reisz, 2011, seems to be only for visual alignment of the typeset matrix’s columns). This approach has also drawn criticism (e.g., see discussion by Kornet & Turner (1999)) because this polymorphic character state is not mutually exclusive with other states, as is the convention for character construction (e.g., Sereno, 2007). There is no consensus among any subset of workers as to the treatment of polymorphisms, but it should be emphasized that whatever approach is being employed should be explicitly stated, and ignoring polymorphisms, as seems evident from many previous studies, should be discouraged.

Conjunctio multidens merits discussion because it is scored for the most polymorphisms in derivates of the Schoch (2012) matrix. The holotype and the referred specimen long referred to as the Rio Arriba Taxon (UCMP 40103) were historically separate OTUs. They are superficially somewhat different (e.g., Schoch & Sues, 2013:fig. 2), which may reflect that the holotype is nearly twice as large. Schoch & Sues (2013) did not report their new composite OTU of these two specimens, so the composite OTU currently in use is that of Liu (2018). There are five polymorphisms for C. multidens: interorbital width (23); tabular process (46); preorbital-postorbital ratio (52); tabular horn (64); and pointed snout (67). Two of these characters (23, 52) are related to skull proportions, which conceivably could be ontogeny-related. Another two (46, 64) relate to the same part of the tabular, and the first of these has been explicitly stated to be ontogeny-related in some dissorophids like Cacops (e.g., Reisz, Schoch & Anderson, 2009). Any workers continuing to use this matrix should consider rescoring these with an eye towards ontogeny, as I did here, since polymorphisms are not scored for taxa represented by a much narrower size range of specimens that capture anatomical variation (e.g., the tabular horn of Cacops morrisi).

The last character (pointed snout) appears to be a typographic error introduced by Holmes, Berman & Anderson (2013). Per Schoch (2012), a pointed (not parabolic or square-shaped) snout is only found in Broiliellus, Dissorophus, and Scapanops (“Admiral Taxon”); he scored it as absent in the holotype and the referred specimen of Conjunctio multidens. However, it is scored as present in the holotype of C. multidens by Holmes, Berman & Anderson (2013). This was only identified by examining their matrix; they did not list it in the text despite listing other scoring changes, further evidence that it is a typographic error. The final line of evidence is that the “Admiral Taxon” was scored as lacking a pointed snout in this derivate, contrary to Schoch. The holotypes of C. multidens and Scapanops neglectus were scored as successive lines and could have been misread (Schoch’s matrix was available only as scoring strings in a typeset figure). I hypothesized that this error could account for the shift of C. multidens from the base of Cacopinae in Schoch & Sues (2013), who presumably scored the composite as lacking this feature (as in Schoch (2018a)), to the base of Dissorophinae in Liu (2018), who scored C. multidens as polymorphic. Since this was a simple test, I corrected the scoring of C. multidens from ‘0 & 1’ to ‘0’ and that of S. neglectus from ‘0’ to ‘1’ and reanalyzed Liu’s matrix. The MPT length increased by one step, but the same number of MPTs were recovered (six), and the strict consensus topology remained unchanged, indicating that this one propagated error is not the sole explanator of Liu’s topology.

Software selection

One of the relatively understudied factors that may produce conflicting topologies between studies is the choice of software. While it is well-known that different programs often produce different topologies, the actual differences are rarely examined. As noted in the Methods, comparisons of previous studies suggest that the choice of PAUP* versus TNT is exerting a meaningful influence on dissorophid topology. Analysis 8 (trematopid-focused; Fig. 17) did not recover any differences in the strict consensus. Conversely, Analysis 9A (original matrix of Dilkes (2020); Fig. 19) identified a few topological differences between strict consensus trees of the restricted taxon sample. TNT recovered more resolution in the parallel of Dilkes’ (2020) original matrix, but all newly recovered nodes are compatible with the equivalent polytomies of the original study (e.g., the resolution of the trichotomy of Broiliellus brevis, Broiliellus olsoni, and Broiliellus texensis; Fig. 19D).

Because some of the inherent differences between programs (e.g., default branch collapsing rule, rounding rule for nodal frequency) do not correlate with biological principles that would clearly support using one parameter over another (compared to a parameter like character ordering), the topology of one program is not more “biologically accurate” or an “overestimate.” These specific programs have purportedly recovered different topologies for the same matrix in other studies (e.g., Schoch, 2013, claimed to have recovered less resolution using TNT), but Marjanović & Laurin (2019:4) have addressed many of these findings and note that they were in fact the result of errors or a non-equivalent search between programs. At least some of these likely relate to poorly documented nuances of various programs (e.g., the need to run a second round of TBR branch-swapping from a stored set of MPTs in TNT in order to obtain all MPTs) or to different heuristic algorithms for relatively large datasets (e.g., the New Technology Search in TNT and the parsimony ratchet (PAUPRat) that is used in tandem with PAUP*). For example, neglecting to run a second round of branch-swapping in TNT could produce more resolution if the first set of MPTs (suboptimal) was only a small subset of the total MPTs. Less-than-best practices, such as running a heuristic search with a relatively low number of replicates (either to identify MPTs or for bootstrapping), could also explain the recovery of an incomplete set of MPTs or suboptimal trees.

Other studies not addressed by Marjanović & Laurin (2019) have reported different topologies recovered by heuristic TNT and PAUP* (e.g., Kurochkin et al., 2011; Han et al., 2016; Audo, Barriel & Charbonnier, 2021), but assessing whether these too might have failed to obtain all MPTs is beyond the scope of this study. A recent comparison of performance of different parsimony programs on phylogenomic data by Goloboff, Catalano & Torres (2021) noted that PAUP* recovered optimal trees in all datasets but one compared to TNT. Other paleontological studies have recovered the same number and length of MPTs between programs, both with large numbers of MPTs (e.g., Spaulding, O’Leary & Gatesy, 2009; Ford & Benson, 2020) and with small numbers of MPTs (e.g., Davesne et al., 2016; Villalobos-Segura, Underwood & Ward, 2021). Most studies do not report the majority of employed parameters, so while it can be reasonably assumed that most studies use the default settings (with various differences between TNT and PAUP*, e.g., branch-collapsing rule, TBR reconnection limit), this is not actually known. These factors also should not result in different results for exact searches, regardless of whether they in fact influence heuristic ones, but many analyses also have taxon samples that exceed the typical computational threshold for exact searches. In this case, it may be that the algorithms for such searches are not in fact the same (‘branch-and-bound’ in PAUP* versus ‘implicit enumeration’ in TNT). Broadly speaking, implicit enumeration is usually considered to be a specific form of a branch-and-bound algorithm for programming problems with variables of a “0−1” nature (e.g., Balas, 1965; Geoffrion, 1969; Davis, Kendrick & Weitzman, 1971; Breu & Burdet, 1974). Further exploration of possible differences between exact search algorithms in these programs is beyond the scope of this study but should be a focus of future studies.

Given that the explanators for differences between programs are not fully documented, it would be preferable if workers would use the same program as the previous iteration of the same foundational matrix to be properly comparative or to restrict any comparisons made with analyses that used a different program. It would also be beneficial if workers would provide the complete set of MPTs in supporting information (as I do here) rather than merely depictions of different consensus trees. This would have the advantage of allowing workers to compute additional consensus topologies not presented in the paper and is a key step towards reproducibility.

Support metrics

In theory, support metrics could be one means of comparing disparate topologies of the same matrix that were recovered from different programs. In Analysis 9A (Fig. 19D), all of the nodes that were recovered in TNT but not in PAUP* had Bremer support of 1 and bootstrap support below 10%. Unfortunately, reporting of support metrics for dissorophid studies is rather uneven (Table 3), which prevents a full comparison of previous studies. Holmes, Berman & Anderson (2013) and Schoch & Sues (2013) did not report any support metric. Maddin et al. (2013) reported bootstrap values, whereas Liu (2018) reported Bremer values. Schoch (2012) and Dilkes (2020) reported both metrics. There are thus two issues: (1) it is not possible to directly compare a node’s Bremer support in one study to the equivalent node’s bootstrap support in another study; and (2) even if the same metric is reported, if the analyses were conducted in different programs, the nodes are not necessarily properly equivalent (e.g., they may not contain the same subset of taxa). This holds especially true for heuristic searches, which may not recover all (or any) MPTs. Therefore, the only proper comparison is between Dilkes’ PAUP* analysis and Schoch’s PAUP* analysis (Table 3), which shows that both Bremer and bootstrap support has declined for dissorophid nodes, sometimes substantially so (e.g., Kamacops + Zygosaurus).

Table 3 Comparison of reported support metrics for focal nodes from dissorophid-focused analyses.

Clade	S12	MFEM13	L18	D20 (A)	This study (A)	D20 (W)	this study (W)	
Olsoniformes	3/100	NR/91	NR	>3/90	>3/75	4/94	4/77	
Trematopidae	2/77	NR/84	NR	>3/98	2/61	4/95	3/63	
Dissorophidae	3/93	NR/82	3/NR	3/86	2/53	3/90	3/58	
Cacopinae	3/98	–	3/NR	1/54	–	2/70	1/39	
Cacops	–	–	1/NR	1/71	1/41	1/73	2/46	
Dissorophinae	1/76	NR/72	3/NR	–	–	1/71	1/50	
Post-Platyhystrix	1/60	NR/69	1/NR	1/51	1/32	1/71	2/53	
Post-Aspidosaurus	1/<50	NR	–	1/45	–	2/72	1/52	
Note:

All nodes are from strict consensus trees except for Schoch (2012). Abbreviations refer to publications: S12, Schoch (2012); MFEM13, Maddin et al. (2013); L18, Liu (2018); D20, Dilkes (2020). ‘A’ and ‘W’ refer to the taxon samples employed by Dilkes: all taxa (A) and without wildcards (W). The same sublettering is used for this study (Analysis 7). An en-dash indicates that a node was not recovered, and ‘NR’ means that the value was not reported. Because the composition of Cacopinae and Dissorophinae sometimes includes certain wildcard taxa (e.g., Conjunctio multidens) or do not include longstanding nominal members, these nodes are restricted in this specific comparison to the same stable constituent taxa in this table. Cacopinae is comprised of Cacops + Anakamacops + Kamacops + Zygosaurus; and Dissorophinae is comprised of Broiliellus + Diploseira + Dissorophus. Bremer decay indices are listed before the forward slash, and bootstrap values are listed after the forward slash.

Reporting of at least one support metric is standard practice in contemporary phylogenetics, so in my opinion, studies without any support metrics should be regarded skeptically (Holmes, Berman & Anderson (2013), and Schoch & Sues (2013), in this context). As aptly put by Sanderson (1995:299), “without some assessment of reliability, a phylogeny has limited value. It may still function as an efficient summary of available information on character-state distributions among taxa […] but it is effectively mute on the evolutionary history of those taxa”. Although expressed more in the context of standard parsimony bootstrapping, this stance is also valid for Bayesian analyses; not reporting posterior probabilities is concerning, especially because posteriors tend to overestimate support (Alfaro, Zoller & Lutzoni, 2003; Cummings et al., 2003; Douady et al., 2003; Erixon et al., 2003; Simmons, Pickett & Miya, 2004; Zander, 2004) and especially with all-clades compatible trees, which force full resolution. This is a shortcoming of the study by Atkins, Reisz & Maddin (2019), which was the previous densest sampling of olsoniforms but which presents only the all-clades compatible tree without posterior probabilities (Figs. 2 and 3 therein).

Aiming to achieve resolution regardless of support, or interpreting topologies without consideration of support, is problematic because it encourages tinkering with the matrix to produce either some semblance of resolution where none previously existed or to produce a topology consistent with previous analyses or with the authors’ preconceived notions. The same is true of dismissing studies that recover poor resolution or that only emphasize well-supported nodes. It is always better to conservatively derive conclusions from only well-supported nodes rather than basing them on poorly supported, possibly spurious relationships. Indeed, methods to penalize spurious relationships (e.g., Rineau, Zaragüeta i Bagils & Laurin, 2018) should be better utilized. A polytomy may be unsatisfactory and is unlikely to depict the actual evolutionary history of a clade, but it is more likely to be correct insofar as it encompasses a genuine clade. This point is salient here because nodes of the in-groups of this study’s analyses tended to be poorly supported except in relatively restricted analyses. Some of this is clearly associated with the sampling of poorly known taxa, reflected in the weakly resolved strict consensus topologies (e.g., Figs. 6, 9, 10). However, it bears noting that in previous studies of both dissorophids and trematopids, most dissorophid nodes fail to meet the threshold for meaningful support for at least one metric (Tables 3, 4). My identification of widespread “assumed” scores also questions the robusticity of previously recovered nodes (like the falsely homogenous Cacops).

One other point to emphasize is that bootstrap support in any TNT analysis is surprisingly low–often below 50%–even for many nodes that have strong Bremer support, which is hardly unique to this study (see Schoch, 2013, and other studies cited below). Many of these nodes are for major clades, like Dissorophidae (Figs. 8B, 10B, 12, 15B, 15D), and some nodes are not even recovered in the bootstrap tree (<1% of bootstrap replicates; Figs. 10, 19D). I have personally never run a PAUP* analysis in which a node recovered in the strict consensus was not recovered in the bootstrap tree, but this occurred in some analyses of the much larger dataset of Marjanović & Laurin (2019:figs. 10, 11, 14, 18, 19) and presumably could occur in other studies that recover poorly resolved strict consensus trees.

Conversely, some nodes that were not recovered in the strict consensus topologies of this study were recovered in a small (<20%) of bootstrap replicates; these tended to be historical relationships (e.g., Acheloma cumminsi + Phonerpeton pricei). This may relate to limitations of TNT, which does not allow the user to define additional parameters of the heuristic search when bootstrapping (presumably TNT uses its default heuristic search parameters: 10 replicates, holding 10 trees per replication, and with TBR), in contrast to PAUP*. The weak support of these nodes underscores the essentiality of reporting support metrics alongside the topology. A cursory survey of recent temnospondyl studies that used TNT reveals three main clusters: (1) studies that do not report any support metrics (e.g., Liu, 2016; Schoch, 2018b, 2019; Schoch & Voigt, 2019; Schoch, Henrici & Hook, 2020; Schoch, Werneburg & Voigt, 2020; Schoch & Milner, 2021); (2) studies that only report Bremer support (e.g., Marsicano et al., 2017; Schoch, 2018a); and (3) studies that report the same pattern of weak bootstrap support within the in-group, even for nodes with very strong Bremer support (e.g., Eltink et al., 2016; Eltink, Da-Rosa & Dias-da-Silva, 2017; Marzola et al., 2017; Pacheco et al., 2017; Chakravorti & Sengupta, 2018; Eltink, Schoch & Langer, 2019).

One source of comparatively low(er) bootstrap values could be a default setting in TNT. Bootstrapping in TNT displays frequency differences (Group present/Contradicted (GC)) by default, not absolute frequencies, following Goloboff et al.’s (2003) preference for using GC frequencies to assess support. Because TNT is only available as command-line for Mac users, that this is the default may not be readily apparent. However, informal comparisons of a few of my analyses did not recover GC frequencies that were substantially lower (>10%) than the absolute frequencies. One example comparison is provided in Table 5 for Analysis 9A (TNT analysis of Dilkes’ original matrix, without wildcards); the greatest difference between frequencies is 4%, although two nodes dropped below the 50% cutoff for meaningful support. The same pattern of generally minimal difference is noted in other paleontological studies that report both absolute and GC frequencies (e.g., Ezcurra, Scheyer & Butler, 2014; Nesbitt & Ezcurra, 2015; Schultz, Langer & Montefeltro, 2016; Marsh et al., 2019; Agnolin et al., 2020; Scheyer et al., 2020). The temnospondyl studies noted above only reported one type of bootstrap frequency and did not specify which one it was.

Table 5 Comparison of absolute and GC frequencies for nodes recovered in Analysis 9A (taxon sample without wildcards; Fig. 19D).

Node	Absolute	GC	Change	
Olsoniformes	88	87	−1%	
Trematopidae	84	84	–	
Acheloma + Phonerpeton	75	71	−4%	
Anconastes + Tambachia	82	81	−1%	
Acheloma + Phonerpeton + Anconastes + Tambachia	62	60	−2%	
Ecolsonia + Fedexia	51	48	−3%	
Dissorophidae	77	77	–	
Post-Platyhystrix dissorophids	52	51	−1%	
Post-Aspidosaurus dissorophids	52	51	−1%	
Cacopinae	57	54	−3%	
Cacops	53	53	–	
Dissorophinae	51	47	−4%	

Phylogenetic relationships of olsoniforms

A defensible consensus

The above discussion has cast substantial doubts on essentially all facets of olsoniform phylogenetics, ranging from topological differences between studies to substantial errors introduced in the character matrices to disparity between programs and in reporting of support metrics. Clearly there are many more unknowns than resolved quandaries that will require significant work to address. Given this, what can be confidently concluded regarding the phylogenetic relationships of Olsoniformes?

The monophyly of Olsoniformes, Trematopidae, and Dissorophidae is universally recovered and well-supported. The only previous study with appreciable olsoniform sampling that did not recover Olsoniformes was Fröbisch & Reisz (2012); the dissorophid sample (five species) is instead more closely related to the terrestrial amphibamiforms (historical ‘amphibamids’). Limited sampling may also explain why Olsoniformes was not recovered in Marjanović & Laurin’s (2019) analysis of early tetrapods, which only sampled five taxa (Acheloma cumminsi, Broiliellus brevis, Ecolsonia cutlerensis, Phonerpeton pricei, and a composite of Mattauschia laticeps and Mordex calliprepes as “Mordex laticeps”). A lack of monophyly was consistent throughout analyses of the original matrix of Ruta & Coates (2007) to the unaltered reanalysis of this matrix to various other derivates with constraints, updated scores, and the addition of “M. laticeps” (not in the original matrix).

The relationships of trematopids remain poorly resolved regardless of the improved resolution in the new analyses (Figs. 14, 17, 18). The only node that is almost always recovered and with good Bremer and bootstrap support is Acheloma + Phonerpeton (usually A. cumminsi and P. pricei). The pairing of Anconastes vesperus and Tambachia trogallas is also recovered in most studies, both trematopid-focused (Figs. 5, 14, 17) and non-trematopid-focused (Figs. 10, 12B, 15, 19, 20), but it generally has low Bremer and bootstrap support (Table 4). Relationships of other taxa seem to be highly susceptible to sampling of other trematopids, as I previously noted (Gee, 2020b) and as seen in comparing different islands recovered in Analysis 8 (Fig. 18). In analyses that recovered appreciable resolution, Ecolsonia cutlerensis remains a trematopid regardless of whether the sampling focuses on dissorophids, on trematopids, or on olsoniforms in general (Figs. 10, 12, 16, 19, 20). The main exception is Analysis 8 (Fig. 17), in which one MPT recovered the nominal trematopids as a grade (Fig. 18C).

Table 4 Comparison of reported support metrics for focal nodes from trematopid-focused analyses.

Clade	B10	B11	PR11	G20	This study (TNT)	This study (PAUP*)	
Olsoniformes	2/NR	2/NR	2/66	NR/55	3/32	3/58	
Dissorophidae	2/NR	1/NR	NR	NR/97	>5/81	>5/93	
Trematopidae	3/NR	5/NR	5/82	NR	3/57	3/78	
Acheloma + Phonerpeton	4/NR	7/NR	–	NR/92	3/57	3/78	
Anconastes + Tambachia	1/NR	1/NR	6/77	–	1/21	1/42	
Ecolsonia as trematopid?	No	No	Yes	Yes	No	No	
Note:

All nodes are from strict consensus trees. Abbreviations refer to publications: B10, Berman et al. (2010); B11, Berman et al. (2011); PR11, Polley & Reisz (2011); G20, Gee (2020b). Nodal support of Gee (2020b) refers to the analysis of that study that sampled all twelve trematopids at the species-level (Fig. 6 therein). This study’s nodal support is derived from Analyses 6 and 8 (Figs. 14B, 17). An en-dash indicates that a node was not recovered, and ‘NR’ means that the value was not reported. Note that Trematopidae in Analyses 6 and 8 of this study only includes Acheloma cumminsi (the specifier for the clade) and Phonerpeton pricei. Bremer decay indices are listed before the forward slash, and bootstrap values are listed before the forward slash.

The classic concept of dissorophid relationships is a base of Platyhystrix rugosa and Aspidosaurus binasser and two higher nested subfamilies, Cacopinae and Dissorophinae. This is recovered in most analyses of Dilkes’ (2020) matrix or the revised version (Analyses 7, 9; Figs. 15, 19, 20) here, as well as in the mirrored analysis with his taxon sample and my matrix (Analysis 5; Fig. 12B). However, both the post-Platyhystrix and the post-Aspidosaurus nodes are weakly supported (Bremer decay index never higher than 2; bootstrap frequency never higher than 56%). As mentioned above, although P. rugosa is usually recovered as diverging first, the weak nodal support for all post-Platyhystrix dissorophids may be linked to the interpretation of osteoderms (or lack thereof) in P. rugosa (see also Dilkes, 2020:fig. 12B). Cacopinae consists of at least Cacops and usually the three sampled middle Permian dissorophids (Anakamacops, Kamacops, Zygosaurus). However, the intrarelationships remain poorly resolved (e.g., Liu, 2018; Dilkes, 2020; Figs. 4, 15, 19, 20), and the erroneous scorings found in other matrices are not the only confounding factor; the extremely fragmentary nature of Zygosaurus (historically scored the same as Kamacops for all overlapping characters) and the loss of its holotype are probably the more pressing matter. The closer relationship of Cacops morrisi to Cacops aspidephorus than to Cacops woehri that I recovered for the first time is in agreement with qualitative comparisons and diagnoses of these species. The composition of Kamacopini remains unresolved (Figs. 4, 15, 19, 20). Dissorophinae typically includes at least Broiliellus, Diploseira, and Dissorophus (Figs. 3, 4, 7, 8, 10B, 12B, 15D, 19D, 20C). However, the intrarelationships of these taxa remain poorly resolved; there are either large polytomies or very poorly supported resolution. The placement of Brevidorsum profundum, Conjunctio multidens, Scapanops neglectus, and Reiszerpeton renascentis is hardly resolved, and they should only be considered as unplaced dissorophids (contra Schoch & Milner, 2014, who consider all four as cacopines).

A review of the taxonomic composition of Dissorophidae

This section discusses the state of affairs with an eye towards future work. Having previously discussed the state and prospects of trematopid research (Gee, 2020b; see also, Milner, 2018) I now focus on dissorophids, summarizing the present state of knowledge for the four subfamilies (Aspidosaurinae, Cacopinae, Dissorophinae, and Platyhystricinae) and highlighting future areas in need of redress that will hopefully help to refine phylogenetic analyses.

The status of Cacops

Currently, there is a consensus that there are three nominal species of Cacops: C. aspidephorus (type species), C. morrisi, and C. woehri (Fig. 21). For over a century, the cranial morphology of C. aspidephorus was largely unknown, in stark contrast to the recently discovered material of C. morrisi and C. woehri from Richards Spur (Reisz, Schoch & Anderson, 2009; Fröbisch & Reisz, 2012; Fröbisch, Brar & Reisz, 2015; Gee & Reisz, 2018a; Gee, Bevitt & Reisz, 2019). Some previous analyses have surprisingly failed to recover a monophyletic Cacops, and all others could not resolve the interrelationships of the three species (Fig. 21). In fact, Dilkes (2020) is the only study to recover a monophyletic Cacops when all three species were sampled (Fig. 21J); while bootstrap support was strong (>70%), Bremer support was not (1).

Figure 21 Morphological and phylogenetic concepts of Cacops.

(A–C) Reconstruction of the skulls of the three species of Cacops; (D–K) pruned topologies from previous studies with the most exclusive clade that includes all sampled species of Cacops (shaded in green). (A) Cacops aspidephorus (from Anderson, Scott & Reisz, 2020); (B) Cacops morrisi (modified from Reisz, Schoch & Anderson, 2009); (C) Cacops woehri (new); (D) strict consensus of Fröbisch & Reisz (2012); (E) 50%-majority-rule consensus of Schoch (2012); (F) strict consensus of Schoch & Sues (2013); (G) strict consensus (left) and 50% majority-rule consensus (right) of Maddin et al. (2013); (H) strict consensus (left) and 50% majority-rule consensus (right) of Holmes, Berman & Anderson (2013); (I) strict consensus of Liu (2018); (J) strict consensus with wildcard dissorophids included (left) and excluded (right) of Dilkes (2020); (K) strict consensus of Gee et al. (2021). The two identical topologies figured for Holmes, Berman & Anderson represent their two analyses (with and without scoring changes for the “Rio Arriba Taxon”). The topology of Cacopinae is unchanged in their 50%-majority rule consensus trees. Cool colors represent skull roof elements; warm colors represent palatal elements. Scale bars equal to one cm for parts A–C.

The historic results are surprising given the stark dissimilarity of Cacops woehri to the other two species. Anderson, Scott & Reisz (2020) questioned whether C. woehri is properly placed in the genus, a suspicion that I agree with. My matrix is the first to resolve the interrelationships of the three species, with C. woehri as the sister taxon to the pair of Cacops aspidephorus and Cacops morrisi (Figs. 7–11), although this relationship is also recovered with the updated version of Dilkes’ matrix (Fig. 14), which substantially alters scores of this genus in particular. This topology (nor any broadly speaking) cannot differentiate between competing concepts of the genus that seek to ensure monophyly, one in which C. woehri is placed in Cacops and one in which it is placed in a different genus that is closely related to Cacops. The lack of postcrania hinders this discussion but also places an emphasis on the revision of Parioxys ferricolus, which Schoch & Milner (2014) note is similar to Cacops. The few photographs and early descriptions of P. ferricolus (Moustafa, 1955a, 1955b) do not indicate the presence of the tubercular ornamentation found in other cacopines, which could indicate that C. woehri is either closely related to, or synonymous with, P. ferricolus. The latter has long been marginalized in the literature, so it has not usually been compared to dissorophids, including by Fröbisch & Reisz (2012) in naming C. woehri.

The status of Broiliellus

Broiliellus is the most speciose dissorophid genus, with five valid species: B. arroyoensis, B. brevis, B. olsoni, B. reiszi, and B. texensis (Williston, 1914; Carroll, 1964a; DeMar, 1967; Holmes, Berman & Anderson, 2013). However, even with the exclusion of “Broiliellus” hektotopos, the concept of Broiliellus remains convoluted (Fig. 22). In all previous analyses but one, the sampled species of Broiliellus do not form a clade. The one analysis that does recover a clade only sampled two species and three dissorophines in total (Schoch & Sues, 2013; Fig. 22B).

Figure 22 Morphological and phylogenetic concepts of Broiliellus and Dissorophus.

(A–H) Pruned topologies from previous studies with the most exclusive clade that includes all sampled species of Broiliellus and Dissorophus; (I–M) reconstruction of the skulls of species of Broiliellus and Dissorophus with known sutures. (A) 50% majority-rule consensus tree of Schoch (2012); (B) strict consensus tree of Schoch & Sues (2013); (C) strict consensus (left) and 50% majority-rule consensus (right) of Maddin et al. (2013); (D) strict consensus (left) and 50% majority-rule consensus (right) of Holmes, Berman & Anderson (2013) with modified scorings for the “Rio Arriba Taxon” from Schoch (2012); (E) the same, but without modified scorings; (F) strict consensus of Liu (2018); (G) strict consensus with wildcard dissorophids included (left) and excluded (right) of Dilkes (2020); (H) strict consensus of Gee et al. (2021); (I) D. multicinctus (from Schoch, 2012); (J) B. texensis (from Schoch, 2012); (K) B. brevis (from Schoch, 2012); (L) B. olsoni (from Schoch, 2012); (M) B. reiszi (from Holmes, Berman & Anderson, 2013). Cool colors represent skull roof elements; warm colors represent palatal elements. Scale bars equal to one cm for parts I–M.

Perhaps the most outstanding issue is that most species of Broiliellus have also not been (re)described in decades. The type species, B. texensis, has never been revised since Williston’s (1914) original description (but see DeMar, 1966b:fig. 4). Material of Broiliellus olsoni is neither substantial nor well-preserved, and that of Broiliellus arroyoensis has no identifiable cranial sutures (DeMar, 1967). As I previously noted, the descriptions of Broiliellus brevis and Broiliellus texensis are dated, short, and with limited figures (Williston, 1914; Carroll, 1964a). Isolated parts of the anatomy (e.g., LEP) are occasionally revised in comparative discussions (e.g., Bolt, 1974b; Dilkes, 2020), but at least the type species would benefit from a thorough redescription with contemporary photography. There are no modern photographs of any of these species other than a cropped palatal view of B. brevis (Witzmann & Werneburg, 2017:fig. 13B), and as a result, some data exist only as scores in matrices. These are probably based on personal observations that cannot be substantiated or reproduced from the literature alone.

A detailed revision of the entire genus might recover a monophyletic Broiliellus, but this possibility seems unlikely at present. One possibility is that these taxa appear morphologically disparate in part because they are differently sized (i.e., this may be partially confounded by ontogenetic disparity). However, it seems more likely that some of the species warrant placement in novel genera if the goal is to ensure monophyly of Broiliellus, perhaps all of them other than the type species. The present topological instability and lack of resolution does not allow for a confident determination of which taxa warrant reassignment or whether any subset of Broiliellus might form its own clade. Therefore, I refrain from erecting novel genera for the non-type species and recommend the use of quotation marks for these species.

The status of Aspidosaurus

This taxon has a convoluted history and likely represents a wastebasket taxon encompassing a semi-conserved osteoderm morphotype (Schoch & Milner, 2014). Most of the species are represented only by fragmentary isolated postcranial material and cannot even be determined to be valid without a good understanding of axial variation (if such variation exists). With the loss of all material of the type species, Aspidosaurus chiton, the functional representative is Aspidosaurus binasser, which preserves substantial axial variation, at least as interpreted. This discussion addresses this taxon and its peculiar mosaicism as part of this broader discussion of the state of affairs within Dissorophidae.

Aspidosaurus binasser is known only from the holotype, which consists of a partial skull (in several pieces) and numerous osteoderms and vertebral fragments (Berman & Lucas, 2003). Among the postcranial material are three types of osteoderms; the type 2 of Berman & Lucas is the stereotypical Aspidosaurus morphotype, while types 1 and 3 are hyperelongate, ornamented spines similar to those of Platyhystrix rugosa. The material that I show in Fig. 23, collected in the late 19th century from Wichita County, TX, consists of similar spines that lack the tubercles and the curvature of the spine (at least where preserved) that diagnose Platyhystrix. The style of ornamentation and the transverse compression of the spines are thus very similar to the type 1 and type 3 osteoderms in A. binasser. Notably, if previous workers’ conjecture on the Wichita County sites is correct (see Romer, 1928:80; Romer, 1935:1617; Milner & Schoch, 2013:116), the Wichita County localities are in the lower-middle part of the Wichita Group (Nocona-Petrolia Formations), much lower in section than the type locality of A. binasser (Arroyo Formation at the base of the Clear Fork Group). While stratigraphic occurrence is not diagnostic in a taxonomic sense, it is highly informative for dissorophid taxonomy. Despite a continuous record of the clade throughout the Early Permian of Texas, no species is known to extend beyond one formation-level unit.

Figure 23 Photographs of neural spines in the collections of the American Museum of Natural History resembling those attributed to Aspidosaurus binasser.

(A) AMNH FARB 23406 (identified as ?Aspidosaurus, collected from the north fork of the Little Wichita River, TX in 1880); (B) AMNH FARB 23407 (identified as ?Aspidosaurus, collected from unknown locality in the Wichita Basin of TX in 1878); (C) AMNH FARB 23408 in part (identified as Aspidosaurus, collected from unknown locality in the Wichita Basin of TX in 1878; only the piece clearly representing an ornamented spine is shown); (D) AMNH FARB 23409 (identified as Aspidosaurus sp., collected from unknown locality in the Wichita Basin of TX in 1878); (E) AMNH FARB 23410 (identified as Aspidosaurus, collected from Shell Point, Archer Co., TX; Nocona Formation in 1878); (F) AMNH FARB 23411 (not identified, collected from Wichita Co., TX in 1878); (G) AMNH FARB 23412 (identified as Aspidosaurus sp., collected from unknown locality in the Wichita Basin of TX). Identifications are based on associated collections cards; the person(s) who identified them or the date of identification is not indicated for any specimen. Scale bars equal to one cm.

At the time of the description of Aspidosaurus binasser, the type 1 and type 3 osteoderm morphotypes had never been reported from another locality, let alone from another formation. This implicitly strengthened the cranial-postcranial association because it suggested that these types were not like the stereotypical Aspidosaurus morphotype or like Platyhystrix–almost always fragmentary, isolated postcrania with a relatively wide stratigraphic range. Material of these taxa is often found at the same sites as cranial material of other dissorophids or temnospondyls, but there is usually not an assumption that the postcrania pertain to an isolated skull just because there are no duplicated skeletal regions. The new observation that the type 1 and type 3 morphotypes appear much lower in section suggests that these purportedly diagnostic morphotypes might in fact belong to a taxon (or taxa) that has a similarly skewed fossil record.

If there was direct articulation between the cranial and postcranial remains attributed to the holotype of Aspidosaurus binasser, the question could be settled quickly. However, there is no direct articulation between the preserved occiput and any of the postcrania. Berman & Lucas (2003) gave no indication as to the nature of the locality, such as the distribution or association of remains. Their mention of indeterminate synapsid material implies that no other distinct dissorophid was identified. Secondly, despite the preservation of the occiput and enough presacral vertebrae to estimate at least 20 positions, neither the atlas nor axis were identified. Thirdly, there is no direct articulation between any two of the three osteoderm morphotypes; they are either isolated fragments or short blocks with only one type. The same applies to the newly reported material. Lastly, no other dissorophid preserves the same stark variation along the axial column purported for Aspidosaurus binasser (Berman & Lucas, 2003, argued for some variation in Aspidosaurus chiton based on Broili (1904)), but numerous taxa preserve essentially no variation in either osteoderms or vertebrae throughout the presacral column (e.g., Broiliellus, Cacops, Dissorophus). The most substantial variation is either in the curvature of different positions in Platyhystrix rugosa to form the sail (e.g., Lewis & Vaughn, 1965) or the transition from a double to a single series in Diploseira angusta (Dilkes, 2020). The former is not apparent in A. binasser, and the latter is characterized mostly by a change in the number of series and the ventral flanges, not regional hyperelongation as in A. binasser. Therefore, the cranial-postcranial association essentially hinges on the assumption that there is likely only one dissorophid at any given site.

Berman & Lucas (2003:244) indeed argued that “there is no reason to suspect more than one individual is represented.” Prior to 2003, there were almost no localities with more than one named dissorophid (e.g., Coffee Creek/Romer’s locality 34), which supported the assumption that dissorophid-bearing localities preserve only one dissorophid taxon (like the Cacops Bone Bed; Williston, 1910; or the Parioxys bone bed of Moustafa (1952)). Since then, however, the presence of multiple dissorophids at a single locality has been documented, and their skeletal representation can be highly uneven. Richards Spur is an excellent case study; Cacops morrisi is known from abundant cranial and postcranial material, and Cacops woehri is only known from semi-abundant cranial material. In contrast, an indeterminate dissorophine is represented only by a headless skeleton and isolated forelimb material, which I left unnamed and not associated with another taxon represented only by cranial material (C. woehri in this case; Gee & Reisz, 2018b; Gee, Bevitt & Reisz, 2019), and Aspidosaurus is represented by a single pair of articulated osteoderms (Gee, Bevitt & Reisz, 2019). Corn Hill in Archer County, TX, USA; the type locality of Brevidorsum profundum and Reiszerpeton renascentis in Archer County; and the Archer City Bonebed are all additional examples of multi-dissorophid sites. Most dissorophid-bearing sites with only one documented dissorophid are type localities that have not produced much, if any, other tetrapod material. Collectively, these observations further the possibility that the holotype of Aspidosaurus binasser could really be a chimera of two taxa, one represented largely or exclusively by cranial fragments and one represented only by fragmentary postcrania.

I have not been able to examine the holotype of Aspidosaurus binasser myself given the present circumstances, but I doubt that I would be able to identify new evidence either definitively proving (e.g., cranial-postcranial articulation) or definitively disproving the association (e.g., identification of duplicated elements). Therefore, I doubt that a redescription is warranted, which is why I raised these points here. Nonetheless, I believe that there is good reason to suspect that not all of the elements attributed to the holotype of A. binasser belong to either a single individual or to the same taxon. The purportedly diagnostic type 1 and type 3 osteoderms are probably more like the stereotypical Aspidosaurus morphotype: one that persists for long time intervals and which does not constitute a true clade, let alone one species, across its range. I do not rule out that some species could have been more stratigraphically extensive than others, but the appreciable fossil record of dissorophids indicates that morphospecies were short-lived. The scoring of A. binasser is not greatly influenced by the cranial-postcranial association in either my matrix or that of other workers, but it would be preferable to restrict the characterization in the future if characters related to spine hyperelongation or axial variation are introduced. Chimerism of A. binasser would affect the validity of both A. chiton and A. binasser since the latter’s diagnosis is based only on the combination of osteoderm types. I make no nomenclatural acts without having examined the type of A. binasser but highlight these issues since the interpretation has not been previously questioned.

The status of Platyhystrix

There has only ever been one species of Platyhystrix, so it may be surprising to see this taxon discussed. Here I focus on the status of the holotype. AMNH FARB 4785 is a multi-taxic batch of material that was first designated as the holotype of “Aspidosaurus apicalis” (Cope, 1881). That taxon is of dubious validity, but the holotype of Platyhystrix rugosa, extracted from this batch by Case in 1910 (as “Ctenosaurus rugosus”), has retained the same number in the literature despite that number representing two taxa. Apparently, the Platyhystrix component was given a subletter designation (4785a) to differentiate them, per a collections tag with “Ctenosaurus rugosus” written on it, but this differentiated number never appeared in the original description or the subsequent literature. Since at least DeMar (1966b:76), the portion considered to be the holotype of “A. apicalis” (AMNH FARB 4785 proper) was considered lost (e.g., Bolt, 1974a; Berman & Lucas, 2003; Schoch & Milner, 2014).

During a collection visit in October 2017, I came across a specimen labeled as AMNH FARB 4785, without subletter designation (Fig. 24), and it matches the description of the holotype of “Aspidosaurus apicalis” instead of that for the holotype of Platyhystrix rugosa. The material assigned to P. rugosa is stated to be several neural spines, while that assigned to “A. apicalis” is specifically the apices (“summits”) of the neural spines (Cope, 1881; Schoch & Milner, 2014), which we would now recognize as osteoderms associated with the spine. As far as I am aware, AMNH FARB 4785a has never been figured, probably because more complete specimens were figured and subsequently utilized as “proxy holotypes” for P. rugosa (Williston, 1911; Langston, 1953; Carroll, 1964a; Lewis & Vaughn, 1965). Williston described a spine that he compared favorably to Case’s holotype, and his figure (pl. 26.1 therein) is of the stereotypical Platyhystrix morphology. This confirms that Case’s (1911:fig. 15) illustrations of AMNH FARB 4785 represent the part that is properly “A. apicalis,” the same subset that I examined. These are distinctly only osteoderms (and one intercentrum of questionable association), not neural spines, and they are much smaller than 11 cm in length (the listed size of one spine per Case). They are, however, in line with the size range given by Cope (less than 4 cm long and 3.5 cm wide). Therefore, as I mentioned previously (Gee, 2018), the holotype of “A. apicalis” (AMNH FARB 4785) is not lost. Instead, that specimen has been repeatedly mistaken for the holotype of P. rugosa (AMNH FARB 4785a), which is missing. There is no record of AMNH FARB 4785a in the museum database, nor was the specimen identified in the most recent inventory (C. Mehling, pers. comm., 2020). No personally examined specimen in the AMNH collection that was assigned to Platyhystrix, Aspidosaurus, or Zatrachys (these being frequently conflated in the early 20th century) matches the description of the holotype of P. rugosa save for one.

Figure 24 Photographs of the holotype of “Aspidosaurus apicalis” (AMNH FARB 4785).

(A) Presumed osteoderm in dorsal and ventral views; (B) the same in lateral view and in either anterior or posterior view (siding is indeterminate); (C) presumed osteoderm in dorsal and ventral profiles; (D) the same in either anterior or posterior view; (E) osteoderm fused to the tip of the neural spine in anterior and posterior views (siding is indeterminate); (F) the same in dorsal view. Scale bars equal to one cm.

AMNH FARB 11544 is a collection of postcrania with a large number of neural spines (Fig. 25), first described, though mostly unfigured, by Berman, Reisz & Fracasso (1981). These purportedly belong to the same individual as AMNH FARB 11545, the only skull of Platyhystrix. The number of spines designated as the holotype of Platyhystrix rugosa was never specified, but it is inferred that “several” is more than two, and the size of some spines of AMNH FARB 11544 is consistent with the measurements given by Case (1910). Some would certainly have been sufficient for proper comparison by Williston (1911). The collections tag indicates that AMNH FARB 4785(a) was collected by David Baldwin in 1881 from the Cutler Formation of Rio Arriba County, NM, the same formation, collector, and collection date as AMNH FARB 11544. Furthermore, Case (1910: 176) mentioned “fragments of scapulae and limb bones associated with the holotype of P. rugosa are typically pelycosaurian in form,” with the association deriving from his interpretation of the spines of “Ctenosaurus” rugosus as those of a pelycosaur. AMNH FARB 11544 includes three large fragments, one of which is a partial glenoid (Fig. 25D), and one of which is a limb end. These were not described by Berman, Reisz & Fracasso (1981), which implies that they also did not believe these fragments belonged to P. rugosa.

Figure 25 Photographs of postcranial material of Platyhystrix rugosa (AMNH FARB 11544).

(A) Isolated neural spines associated with this individual of P. rugosa; (B) neural arches found in association with the other material figured here; (C) sacral ribs associated with this individual (see Berman, Reisz & Fracasso, 1981); (D) material catalogued under the same number but questionably associated with this taxon; the large fragment on the right may represent a synapsid scapula. Berman, Reisz & Fracasso (1981) conjectured that these postcrania of P. rugosa belonged to the skull that is catalogued as AMNH FARB 11545; the skull was not available at the time of my visit to assess the purported fit between one neural spine and a fragment on AMNH FARB 11545. Scale bars equal to one cm.

In my opinion, it seems quite likely that AMNH FARB 4785a was renumbered as AMNH FARB 11544, but that records of this were either not made or were subsequently lost. If the value of AMNH FARB 11545 as the only skull of Platyhystrix rugosa was not recognized for decades after its collection, that would explain why these two specimens were not described for a century. The postcranial material would have drawn little attention until it was determined that it articulated with the cranial material. However, there are no collection records indicating the transfer of the holotype of P. rugosa to a new number (C. Mehling, pers. comm., 2020). Without a record, their equivalency cannot be confirmed, as it remains possible that AMNH FARB 4785a was simply lost. Therefore, the type status designation remains with AMNH FARB 4785a, and I present the conundrum here in the hopes that perhaps other workers may be able to contribute new information to help resolve this matter.

Other dissorophids

Relevant points of the remaining taxa are collated here. Dissorophus multicinctus, while known from an extensive amount of material, would benefit from a systematic redescription. The osteoderms and vertebrae were described by Dilkes (2009) but were otherwise neglected since DeMar (1968). Similarly, the skull has not been redescribed since DeMar, who figured only one complete skull (MCZ 2122-1). While some studies cite Schoch (2012) for the cranial osteology, Schoch only presented a reconstruction, some of which Dilkes (2020) explicitly disagreed with (e.g., position of the jaw articulation). In addition to numerous specimens that were mentioned but not illustrated by DeMar, a number of specimens have been subsequently mentioned or photographed at a low resolution in a single profile. However, these brief documentations are insufficient for a full characterization of the anatomy (e.g., MCZ 1468; Schoch & Milner, 2014:fig. 37C; MCZ 4170, MCZ 4186, and MCZ 4188; Dilkes, 2020:22).

Most of the wildcard taxa (or taxa not previously sampled) have poor prospects for resolving their relationships without new material (e.g., Aspidosaurus novomexicanus, Brevidorsum profundum, “Broiliellus” arroyoensis). Two very fragmentary taxa can only be inferred to be dissorophids if it is assumed that they are dissorophoids (i.e., if these taxa belong to Dissorophoidea, apomorphies of which they generally lack, they most likely belong to Dissorophidae): Iratusaurus vorax and Nooxobeia gracilis. Neither preserves dissorophid synapomorphies (sensu Schoch & Milner, 2014), although the single series of median osteoderms in N. gracilis is suggestive of dissorophid affinities as Olson (1972) proposed (Gee, Scott & Reisz, 2018). It seems doubtful that N. gracilis would represent a chroniosuchian, another tetrapod clade with median osteoderms, as Permian representatives of this group are mostly known from Russia and China (e.g., Golubev, 1998a, 1998b, 1999; Jiang, Ji & Mo, 2017; Liu & Abdala, 2017; Liu, 2020). However, chroniosuchian material is rare and fragmentary, and most records come from the Middle and Late Permian, intervals from which there is little to no record of terrestrial tetrapods in North America (e.g., Lucas, 2001, 2002, 2005, 2013; Reisz & Laurin, 2001; 2002; Lozovsky, 2005; Benton, 2012; Benton, 2013; Olroyd & Sidor, 2017; Brocklehurst, 2020). Recent studies have expanded their range, including to the Upper Permian of Germany (Witzmann et al., 2019), where a single osteoderm-bearing vertebra ascribed to an indeterminate dissorophid was reported from slightly older deposits (Witzmann, 2005). Iratusaurus vorax does not appear like any other temnospondyl with a closed otic notch (e.g., capitosaurs), but its description by Gubin (1980) was extremely cursory, and the material is extremely fragmentary.

Finally, Parioxys bolli may not belong to Parioxys, regardless of the relationship of Parioxys ferricolus to Cacops. The ilium of P. bolli indicates olsoniform affinities, but the remainder of the known skeleton is uninformative; the two sacral ribs that Carroll (1964b) emphasized are not a dissorophid or an olsoniform synapomorphy. The limbs are relatively long, more like those of dissorophids, but ring-like intercentra are a feature found only in Ecolsonia cutlerensis and nearly so in Acheloma cumminsi (Olson, 1941; Berman, Reisz & Eberth, 1985; Dilkes & Reisz, 1987). Of note are lateral projections from each side of the base of the neural arch; these are otherwise found only in E. cutlerensis and in the type 1 vertebrae attributed to Aspidosaurus binasser (Berman, Reisz & Eberth, 1985; Berman & Lucas, 2003). This taxon may well prove to be a trematopid.

Ontogenetic disparity

Although I did not exhaustively test whether ontogenetic disparity might confound or bias the phylogenetic inference of dissorophids, this remains an open question in light of the size disparity across the clade. Within Dissorophidae, this disparity is essentially an order of magnitude, greater than that observed for trematopids (Fig. 26). The temporal distribution of sizes is also non-random, as it was for trematopids. With the latter group, the earliest appearing taxon, Mattauschia laticeps, reached a skull length comparable to that of the much later appearing Ecolsonia cutlerensis (Milner, 2018), but there are other taxa between or concurrent with these occurrences that are represented by smaller individuals. In dissorophids, all of the Middle Permian taxa had skulls with a length of at least 18 cm (there are not even any individual specimens of an inferred smaller size), whereas most Early Permian taxa did not exceed 12–13 cm. Only two, Aspidosaurus binasser and Platyhystrix rugosa, exceed this (Cacops aspidephorus may be a third depending on whether the reidentification of “Trematopsis seltini” to the species level by Milner, 1985, can be substantiated). Therefore, it is possible that dissorophids did increase in size in the late stages of their evolution, perhaps correlated with the extirpation of trematopids and other large-bodied temnospondyls thought to be capable of terrestrial locomotion like edopoids and eryopoids (note that the degree of terrestriality remains contentious for many clades; e.g., Pawley & Warren, 2006; Sanchez et al., 2010; Fortuny et al., 2011; Quemeneur, de Buffrenil & Laurin, 2013; Carter et al., 2021). Whether increased dissorophid size would be a driver or a product of other clades’ extinction is unclear.

Figure 26 Comparative plot of known skull lengths and size ranges of olsoniforms.

Refer to Appendix 9 and Supplemental Table 3 for dataset and methods used to collect measurement data.

While Aspidosaurus binasser and Platyhystrix rugosa are traditionally recovered as the earliest diverging dissorophids (suggesting that large size could characterize most taxa, but that the majority are represented only by juveniles; Gee, 2020a), it is important to note the extremely poor Carboniferous record of dissorophids. Quite possibly, the true earliest diverging dissorophids remain to be discovered. A third hypothesis is that only certain clades of dissorophids achieved large sizes. It is conspicuous that no dissorophine, including the well-sampled Dissorophus multicinctus, exceeded a skull length of 13 cm, while the other three subfamilies did. Size disparity among dissorophids could also relate to ecological differences from trematopids. In contrast to trematopids, for which there are only two localities in Europe (Nýřany, Bromacker) where multiple taxa co-occur, there are many localities where several dissorophids co-occur. Along with anatomical differences such as tooth count, skull proportions, and osteoderm morphology, size differences could also be predicted as an aspect of niche partitioning.

The one analysis that I ran to assess whether ontogenetic disparity might confound dissorophid phylogeny (Analysis 4; Fig. 11) did not recover any clear signals of directional bias. No taxon sampled at the specimen level is recovered as a clade, but most specimens are simply single branches in a cacopine polytomy. Conjunctio multidens is the only taxon in which OTUs are recovered in different positions, and in this case, the large holotype diverges first. On one hand, this taxon’s OTUs contradict one prediction of ontogenetic disparity (stemward slippage of smaller, more immature specimens due to a higher number of what present as “retained” plesiomorphies). Conversely, in the context of olsoniforms, the early diverging position of the largest specimen may still indicate support for an influence of ontogenetic disparity, as the smallest specimens cluster away from the large trematopids. A lack of skeletal overlap seems to produce the pattern of Cacops woehri, in which the holotype (partial skull) and one referred specimen (BMRP 2007.3.5, partial posterior skull) cluster even though they belonged to disparately sized individuals. Specimens of Anakamacops petrolicus have essentially no skeletal overlap, and an ontogenetic range was not sampled for Cacops aspidephorus.

In short, there remain many unknowns and confounding factors that limit the study of size patterns in dissorophids. In an unpublished chapter of my dissertation (Gee, 2020a:388–394), I suggested that niche partitioning between life stages of a given taxon could result in a skewed sample that biases interpretations of “adult” size. For example, numerous skulls of Cacops between 10 and 12 cm in length are known and have thus been dubbed “adults” under a presumption of relative maturity (e.g., Reisz, Schoch & Anderson, 2009; Gee & Reisz, 2018a). In fact, this size range does not come close to approximating the maximum size of Cacops, which could have been nearly double that size based on the single specimen of “Trematopsis seltini,” estimated to 22 cm (=Cacops cf. C. aspidephorus; Milner, 1985). Isolated postcranial remains from Richards Spur suggest that at least one of Cacops morrisi and Cacops woehri also reached a larger size than is reflected by the cranial remains (Sullivan, Reisz & May, 2000; Gee, Bevitt & Reisz, 2019; Gee, 2020a). Intraspecific niche partitioning has not been previously suggested in olsoniforms, but it offers one explanation for the skewed record of even well-sampled taxa like Cacops and for the size disparity between dissorophids if this partitioning extended to physical habitat occupancy. As with trematopids, size evolution in dissorophids remains a quandary that can likely only be resolved with additional collection, although a survey of existing collections might identify outlier datapoints (probably isolated postcrania or fragmentary cranial remains) that document larger body size than traditional proxies (e.g., complete skulls).

Considerations in backbone selection

With increasing computational abilities, paleontologists can sample broad taxonomic swaths while maintaining appreciable in-group sampling of any given clade. Technological advances have also expanded the range of analyses that can be conducted, leading to a proliferation of “big data” studies addressing macroevolutionary questions on scales that were previously infeasible. Most of these studies are phylogenetically informed by an underlying backbone, the selection and design of which is obviously of great import but which is not always rationalized or explained in detail. This final section provides some preliminary comments on temnospondyl backbones in light of this study’s findings.

The most widely utilized topology of Temnospondyli is the computer-assisted supertree of Ruta et al. (2007), which has been incorporated into numerous studies, usually in concert with other tetrapod (super)trees to form a larger informal supertree (e.g., Fortuny et al., 2011; Soul & Friedman, 2017; Dunne et al., 2018; Carter et al., 2021; Dickson et al., 2021). It is noteworthy that this particular topology remains popular among non-taxonomic specialists, whereas temnospondyl workers tend to opt for a variety of alternative backbones. For example, Angielczyk & Ruta (2012) manually modified the topology of Ruta et al. (2007); Witzmann (2013) and Witzmann & Werneburg (2017) used the topology of Schoch’s (2013) non-supertree analysis; Tarailo (2018) used Schoch (2013) as the large-scale backbone, with additions from Ruta et al. (2007) and Marsicano et al. (2017) for small-scale resolution; Witzmann & Ruta (2018) and Pérez-Ben, Báez & Schoch (2019) manually modified the topology of Schoch (2013); and Pardo et al. (2019) and Ruta et al. (2019) used the topology of Pardo, Small & Huttenlocker (2017).

Ruta et al.’s (2007) supertree may remain appealing in spite of its datedness because it is fully resolved and includes numerous wildcard taxa that are rarely sampled in other studies and that are highly unstable when they are sampled (e.g., Bashkirosaurus, Capetus, Collidosuchus, Kashmirosaurus, Lapillopsis, Lysipterygium, Palatinerpeton, Parioxys, Peltobatrachus, Sassenisaurus, Stegops). Of course, the fact that these taxa are excluded from analyses reflects the continued uncertainty over their placement, even in a phenetic framework, but this may only be well-known among taxonomic specialists. Some of these taxa are recovered in the proper clade in Ruta et al.’s (2007) supertree, but with uncertain relationships to other in-group taxa (e.g., Collidosuchus), while others remain of uncertain placement in general (e.g., Lapillopsis). The latter are of greater concern because there is a higher likelihood that their positions in the Ruta et al. topology are spurious.

In general, this topology, while consistent in broad strokes with more recent non-supertree analyses (e.g., Schoch, 2013; Pardo, Small & Huttenlocker, 2017; Eltink, Schoch & Langer, 2019), differs markedly in some areas. Specifically for dissorophoids, branchiosaurids are accepted as nesting within the historical ‘Amphibamidae’ rather than as its sister group (e.g., Schoch & Milner, 2008; Fröbisch & Schoch, 2009); micromelerpetids are a clade at the base of Dissorophoidea rather than a grade of early-diverging branchiosaurids (e.g., Schoch, 2018a); trematopids and dissorophids are sister taxa, not successively diverging branches within Dissorophoidea (e.g., Anderson et al., 2008b); Ecolsonia is a trematopid, not a dissorophid (Polley & Reisz, 2011; Schoch, 2018a; Gee, 2020b; this study); and Parioxys is probably a dissorophid, not an eryopoid (Schoch & Milner, 2014). The taxon sample is naturally outdated as well, but this manifests as what appears to be uneven sampling based on the present body of recognized taxa. Olsoniforms are among the undersampled clades, with only six nominal trematopids and five nominal dissorophids.

These points are not meant as a criticism of the original Ruta et al. (2007) study but rather evidence the predictable datedness after nearly two decades of anatomical and phylogenetic work. Nonetheless, it is clear that Ruta et al.’s (2007) supertree is no longer an accurate reflection of the consensus of temnospondyl relationships and should not be employed as such. Pardo et al. (2019) commented on potential issues of supertree construction and pseudoreplication, especially in light of newer non-supertree analyses performed by taxonomic specialists that challenge historical paradigms and that frequently contradict widely used, but more dated, supertrees. I endorse these authors’ approach to informal supertree construction (p. 11 of their Supplemental File) in which they collate non-supertree topologies recovered by studies whose primary aim was to assess the phylogenetic relationships of a clade and in which they allow taxa with unresolved relationships (or that have never been included in an analysis) to be placed in a polytomy. Such an approach is preferable to enforcing resolution of dubious nature simply in order to achieve full resolution. I encourage non-specialists to consult with relevant phylogenetic/systematic experts with respect to the construction or selection of a backbone. Temnospondyli, like most other speciose clades, continues to be recovered with major areas of instability or weak support. This instability underscores the continued import not only of phylogenetic method refinement and analysis but also of the primary data collection (e.g., fieldwork, descriptive anatomy) that underpins the analysis.

Conclusions

Originally, I had intended to focus this study on expanding my character and taxon sample to broadly represent dissorophids, which I hoped would improve the resolution for trematopids as well. In the process of assessing explanators for topological differences, some of them substantial and often related more to differing degrees of resolution than to drastically different positions of taxa, this study shifted towards a focus on reproducibility and robusticity of previous topologies. This endeavor admittedly became much more exhaustive (and exhausting) than even I had anticipated, and like my trematopid analysis (Gee, 2020b), seems to have identified far more issues than it has resolved. In the end, this study has demonstrated that the phylogeny of Dissorophidae is not resolved, reproducible, or robust. Other key conclusions are outlined below: The widely propagated matrix of Schoch (2012) contains substantial scoring errors that appear to represent “assumed” scores; these scores are for characters where the entire feature is not even preserved, let alone sufficient to be assessed (e.g., postcrania of Cacops woehri). These are unequivocally unfounded and should be regarded as erroneous unless future studies prove otherwise. Almost all of these originated early in the propagation of this matrix and have thus been carried forward into essentially every dissorophid analysis. It is possible that either new material or simply better documentation of existing material might validate these assumptions, but at present, they have no reproducible basis. There are also numerous scores for taxa where a complete element is required to score a character, but none is available for a given taxon (e.g., characters related to skull length for “Broiliellus” olsoni). Given the extensive number of unequivocal errors, missing scores, and unfounded scores (Appendix 5), previous topologies should be treated skeptically, especially with respect to weakly supported nodes, as the corrected matrix recovers an overall less resolved topology.

The use of different programs and variable reporting of support metrics confounds proper comparisons between studies, but these are not the only factors that result in drastically different topologies from studies that are using largely identical character matrices. Persistent wildcards, character construction, and character scoring clearly exert strong influences as well, and a few changes to the matrix can result in drastic changes to the resultant topology. Workers should test for the effects of these phenomena (e.g., analyses with and without wildcard taxa and consensus trees with and without wildcard taxa) and clearly state and justify their preferred approaches.

The intrarelationships of both Dissorophidae and Trematopidae can be resolved through selective taxon sampling, but most in-group nodes fail to meet the thresholds to be considered as “well-supported” for at least one metric: Bremer decay index (>2) or bootstrapping (>50%). Support metrics are one means of comparing topologies produced by different studies, and topologies that are reported without support metrics or with weak support should be treated skeptically.

The only “consensus” relationships within Dissorophidae are the early-diverging position of Aspidosaurus binasser and Platyhystrix rugosa; a Cacopinae that includes Cacops and probably Anakamacops, Kamacops, and Zygosaurus; and a Dissorophinae that includes Broiliellus, Diploseira, and Dissorophus. While the interrelationships of cacopines can be further resolved with some confidence, those of dissorophines cannot at present. Any worker seeking a topology for a backbone in a quantitative analysis should place all other taxa in a polytomy either above or with As. binasser, rather than selecting one of the many different resolved topologies that lack strong support for most nodes.

Supplemental Information

Supplemental Information 1 Appendices 1–3, 5, 8.

Click here for additional data file.

Supplemental Information 2 Appendix 4. Updated matrix of Gee (2020).

NEXUS (.nex) character matrix with the updated scorings for the character matrix of Gee (2020), editable with standard phylogenetic software (e.g., Mesquite, PAUP*).

Click here for additional data file.

Supplemental Information 3 Appendix 6. Updated matrix of Dilkes (2020).

NEXUS (.nex) character matrix with the updated scorings for the character matrix of Dilkes (2020), editable with standard phylogenetic software (e.g., Mesquite, PAUP*).

Click here for additional data file.

Supplemental Information 4 Appendix 7. ZIP file of associated MPTs for each permutation as .tre files.

Click here for additional data file.

I thank the many collections managers and curators who granted me access to their dissorophoid specimens, either first-hand or through loans to my doctoral advisor, Robert Reisz: Mark Norell and Carl Mehling (American Museum); Dave Berman and Amy Henrici (Carnegie Museum); Ken Angielczyk, Bill Simpson, and Adrienne Stroupe (Field Museum); Chris Beard and Dave Burnham (Kansas University Museum of Natural History); Rich Cifelli and Jennifer Larsen (Sam Noble Museum); and Pat Holroyd (University of California Museum of Paleontology). Thanks to Carl Mehling for assistance in trying to elucidate the history of the holotype of Platyhystrix rugosa. Jason Anderson (University of Calgary) also kindly allowed me to examine the material of Cacops aspidephorus on loan to him and transferred the loan of the material of “Fayella chickashaensis” and what is now Nooxobeia gracilis (and took me out to lunch) when I visited his lab in the fall of 2017. Thanks to Jason Anderson, Dave Berman, Adam Huttenlocker, Hillary Maddin, Arjan Mann, David Marjanović, and Jason Pardo for discussions. Thanks to Michel Laurin, David Marjanović, two anonymous reviewers, and the editor, John Hutchinson, for constructive feedback that greatly improved this manuscript. TNT is graciously provided freely by the Willi Hennig Society.

Institutional abbreviations

AMNH FARB American Museum of Natural History, Fossil Amphibians, Reptiles, and Birds, New York, NY

BMRP Burpee Museum of Natural History, Rockford, IL

CM Carnegie Museum, Pittsburgh, PA

FMNH Field Museum of Natural History, Chicago, IL

MCZ Museum of Comparative Zoology, Cambridge, MA

OMNH Sam Noble Oklahoma Museum of Natural History, Norman, OK

ROMVP Royal Ontario Museum, Vertebrate Paleontology, Toronto, Canada

UCMP University of California Museum of Paleontology, Berkeley, CA

Additional Information and Declarations

Competing Interests

Author Contributions

Data Availability

The author declares that they have no competing interests.

Bryan M. Gee conceived and designed the experiments, performed the experiments, analyzed the data, prepared figures and/or tables, authored or reviewed drafts of the paper, and approved the final draft.

The following information was supplied regarding data availability:

The phylogenetic matrices and the recovered sets of MPTs are available as Supplemental Files.

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
