# Peer review of "Returning to the roots: resolution, reproducibility, and robusticity in the phylogenetic inference of Dissorophidae (Amphibia: Temnospondyli)"

_PeerJ, doi:10.7717/peerj.12423_

## Round 0.1 · original submission · Major Revisions

Apologies for the delay in obtaining reviews. This turned out to be a problematic manuscript for us to handle as we had rather disparate reviews and some uncertainty about conflicts of interest, and a change of handling editor was required, and a fourth reviewer who provided quality control and overall judged that the manuscript was strong enough.

However, in the end, the reviews are nicely supportive and constructive, requesting moderate revisions. I concur that a mostly literature-based analysis in the time of COVID-19 is certainly permissible, and the point of the literature is to allow this too, so this is not a fatal flaw of the study. Papers by Simões et al. (2017), Laurin and Piñeiro (2017 and 2018) cover some similar ground and should be better cited herein, to spread the credit around and frame the study in the broader context recommended by reviewer 1 ("data quality"). Further details are provided in the reviews by R1+R2. Please respond to all 4 reviewers' points individually in the body of your revised MS itself, and in the Rebuttal. Some reviewer(s) will need to have a second look at the MS to see if they are satisfied. Thank you.

·

Basic reporting

Mostly good. The paper and literature coverage could be expanded to make the paper relevant outside the very small circle of dissorophoid experts. See the section General comments to the authors for details.

Experimental design

Again, mostly good. I have some suggestions to improve tree comparisons, especially when they were performed with different programs, which may report difference in lengths for the same tree because, among other things, of differences in treatment of polymorphism. See the section General comments to the authors for details.

Validity of the findings

Results seem valid.

Additional comments

The draft by Gee is generally well-written and methodologically mostly sound (see some minor reservations below that probably require minor new analyses). I have not spotted technical, factual errors, but I am not a dissorophoid expert and I have not scrutinized the data matrices or the changes in character scores. I have annotated profusely the pdf file, which should be considered my main report (please make sure that Gee gets it), so here, I will only summarize the main key points.

The main thrust of the paper is that data matrices have often (read « nearly always ») been sloppily coded, with many mis-scores, including obvious shortcomings consisting in scores for characters for which the relevant part of the skeleton is not preserved in the relevant taxa. I agree entirely with him that this is totally unacceptable, and I find distressing that the author, reviewers, and editors of the original study did not spot them, and worse, that several colleagues have re-used the matrix (sometimes more than once) without spotting the problem (and again, neither reviewers nor editor spotted these mistakes). So, this main point is important to make. However, the author frames it in the very specialized perspective of dissorophoid phylogeny. There may be a dozen paleontologists worldwide who work regularly on this group, so this makes the paper relevant to a tiny community. Thus, I recommend that that author frames his findings in the bigger problem of putting more emphasis on data quality rather than data quantity. This concerns all systematists doing phylogenetics using phenotypic characters (mostly paleontologists now, but conceivably some working on the morphology, ecology and behavior of extant taxa too). Recent discussions of these issues can be found in Simões et al. (2017) and Laurin & Piñeiro (2018) and references cited therein. Framing the findings in such a broader perspective can be done in a sentence or two (or a paragraph), but it would greatly enhance the potential impact (and citation potential) of the paper, something that a young scientist looking for a tenure-track position ought to be preoccupied about.

The discussion of the nomenclatural (which he calls “taxonomic”) consequences of this work could be improved by reformulating the text to better reflect the fact that taxa are inherently undelimited in rank-based nomenclature and in fact, are not even required by the zoological code to be monophyletic. Thus, stating that (l 1206) “Cacops comprises three species.” is not entirely accurate. He means that we currently recognize three species. Surely, more were named at some point and put into subjective synonymy. As the term implies, this is a subjective decision that could be wrong. Also, there were probably other lineages in the clade that matches more or less the currently-known Cacops clade, so one day we may discover more of them. For a discussion of such points, which are also relevent elsewhere in the draft (l. 1220-1221; 1252; 1335; 1342), I suggest looking at Laurin (2008), though there is probably no need to cite it in the paper.

I believe that there is an unjustified inference on line 311, where the author indicates that “the higher character count of my matrix (44% more characters) made a branch-and-bound search intractable in PAUP*”. Search time increases linearly with character number, but exponentially with taxon number. Scores can affect search times too, especially with branch and bound, because fewer shortcuts can be taken if there is much homoplasy. So, the issue is certainly not character number, but there is nothing anomalous with the fact that he could not do branch-and-bound with that matrix.

On lines 619-621 and much more importantly 666-668, 1080-1090 and in Permutation 10, the author should verify if the trees of different lengths obtained by the different programs (TNT vs PAUP) reflect differences in calculation by the programs. I know that their treatment of polymorphism, among other things, varies, and ordering scheme can also make a difference. The proper way to do this is to import all trees into the same program and verify their length (I typically do this in Mesquite, which is very user-friendly). Because if one set of trees is shorter than a second set, the first one is preferable, whether their strict consensus is more or less resolved. Yet, this point is not addressed in the paper.

On lines 972-973, the author states that “A lack of resolution does not necessarily indicate that the matrix is compromised or poorly constructed”. I agree! This point could be strengthened by being explained in more detail. Irresolution is better than wrong resolution because the former gives a correct idea of uncertainty whereas the latter looks deceptively satisfactory. In fact, in our (mostly but not entirely simulation-based) work to assess the performance of phylogenetic methods, we penalized methods that yield wrong clades. About this, see Rineau et al. (2018: 19), where we present a mathematical formula to do this quantitatively using a metric derived from 3ia (other ways to do this exist and more could be developed). This point deserves a more extensive treatment because many systematist consider polytomies a failure and some are even tempted to “tinker” with their matrix to “solve the problem”, as the author points out.
This point is relevant on l. 1123 too.

On l. 1134-1135, the author states: “I have personally never run a PAUP* analysis in which a node recovered in the strict consensus was not recovered in the bootstrap tree.” That may well be true for the author's experience, but such differences do occur, including in Marjanovic & Laurin (2019), which he already cites (but not for this point). Compare their figures 10 and 11 (classical MP analysis of revised matrix from Ruta & Coates 2007, but with corrected scores) with the bootstrap analysis (B1, fig. 18). Note multiple differences, such as the placement of close relatives of Lissaphibia (there are probably some in temnospondyls took but these are complex with two sets of topologies depending on the position of anthracosaurs).

On l. 1162-1164, Gee comment again on Marjanovic & Laurin (2019) about their recovered paraphyly of Olsoniformes. Gee does not seem to realize that this paper presents many analyses, including some of the unmodified data by Ruta, some with scores corrected, some with an expanded taxonomic sampling, and the latter seem to refute (to an extant) Gee's suggestion that limited sampling explains this result. I suspect that the problem may be in scores in Ruta & Coates' (2007) matrix that Marjanovic (who did the bulk of the re-scoring) did not find when updating it. But this too is speculation. Perhaps Olsoniformes are paraphyletic? After all, we don't have the true tree of temnospondyls, as Gee himself emphasizes. See my more detailed comment on the annotated pdf file.

References:

Laurin M. 2008. The splendid isolation of biological nomenclature. Zoologica Scr 37:223–233. 10.1111/j.1463-6409.2007.00318.x

Laurin M, and Piñeiro G. 2018. Response: Commentary: A Reassessment of the Taxonomic Position of Mesosaurs, and a Surprising Phylogeny of Early Amniotes. Frontiers in Earth Science 6:1-9. 10.3389/feart.2018.00220

Rineau V, Zaragüeta I Bagils R, and Laurin M. 2018. Impact of errors on cladistic inference: simulation-based comparison between parsimony and three-taxon analysis. Contributions to Zoology 87:25-40.

Simões TR, Caldwell MW, Palci A, and Nydam RL. 2017. Giant taxon‐character matrices: quality of character constructions remains critical regardless of size. Cladistics 33:198-219. 10.1111/cla.12163

·

Basic reporting

All good, except that the style is overwrought in places, making some passages difficult to read. See the attached file for details.

Experimental design

Good, but see the attached file for details.

Validity of the findings

Very good, see attached file.

Additional comments

My review is over 10% as long as the ms, and given that the ms has 142 pages, I provide my review as the attached file. I ask for "major revisions" only because I would like to see the next version; otherwise, I would not call the revisions I ask for "major", even though they are rather numerous.

Reviewer 3 ·

Basic reporting

The study by Gee represents an in-depth treatment of an important and complex clade of dissorophoid temnospondyls, the Family Dissorophidae. It builds extensively on the author's previous experience and excellent puplication record, revealing a fastidious eye for anatomical detail and a proficient use of analytical protocols. Much work remains to be done on this clade, but no stone is left untrurned in this work. Aside from providing a compendium of the history of research and revisions of dissorophids, this study also provides a much needed cladistic framework that could be redeployed in future analyses. The reference list is one of the best I have ever seen, up-to-date and comprehensive. It miswritten clearly and concisely, so the length is justified as this is a real tour de force through an elaborate topic. Figures are clear and essential, with more than adequate captions. The results and conclusions are amply justified.

Experimental design

The research is original and truly important and will be a useful primer for specialists in he field and anyone with interest in early tetrapods (in general) and temnospondyls (in particular) evolution. It demonstrates high standard of execution and fills many gaps in knowledge through attention to morphology and accurate revision, as far as I can tell, of selected characters and coding. Methodology is exhaustive and well written.

Validity of the findings

The findings are intriguing. Loss of resolution in some experiments is expected, but this is not a flaw. All results can be reproduced very easily, given the abundant supplementary material. There are no areas of wild speculation and all statements are justified.

Additional comments

I congratulate the authors on a masterclass delivery

Reviewer 4 ·

Basic reporting

I read this manuscript with great interest. Besides presenting a new phylogenetic analysis of Dissorophidae, the author critically assesses previous phylogenetic studies of the group.

I think this is an important manuscript as it presents a list of shortcomings of previous works that are also applicable beyond the study of Dissorophidae. Crucially, it highlights the importance of documenting and justifying the methodological decisions taken when scoring matrices and running the analyses, and of reporting the results in detail (e.g., reporting support metrics).

Even though there are other researchers in the field who might not agree with some of the statements made by the author (either based on personal opinions or due to the access to further data such as specimens not studied first-hand by Dr. Gee), the level of detail given throughout the manuscript paves the way for future discussions (e.g., in response papers).

Experimental design

- The study is mostly literature-based. This might be a shortcoming as first-hand observations of fossil material are more informative (when conducted by someone familiar with the taxonomic group). However, in this case as the scoring is based on the literature, it reinforces one of the key issues addressed in the manuscript: replicability. On the one hand, literature is more accessible than visiting collections in multiple countries; on the other hand, published and well supported descriptions are more transparent than non-published/non-photographed first-hand observations made by a researcher when scoring.

- I find the narrative structure of the manuscript particularly informative. It is useful to see how the overall goal of the project changed after the first results, and how the author attempted to reconstruct and reproduce previous studies, detailing the limitations he encountered. This is common practice in research, however it is rarely discussed in papers.

- The taxon and character sampling, and the phylogenetic analyses are explained in detail.

Validity of the findings

- The findings and conclusions are well supported by the numerous permutations and the extensive bibliographic work.

Additional comments

- In my opinion, by his thorough revision, the author gives due credit to the previous key dissorophid studies. Instead of simply dismissing or underrating them due to what he thinks is the lack of best practices, he tries to unravel the differences among their results. In this regard, he explicitly states that he is not advocating against propagating an existing matrix (l. 843-844) and shows (Fig. 3) that even his previous works were also derived from Schoch’s matrices. However, I think the manuscript would be improved by including a paragraph explicitly explaining the relevance of the key works the author revises.

- The general tone in which the author criticizes previous studies is appropriate. However:
1. In l. 959-977, where it is stated that inferred character states are not valid, references are needed, or it needs to be made clear that this is the author's opinion. It would also be worthwhile to acknowledge that it was common practice to use “assumed” states, and was not something just done by Schoch.
2. Abstract: “others relate to best practices (or the lack thereof)” may sound pretentious. I would suggest rewording it (e.g., “others relate to discrepancies with respect to what are currently considered best practices”).

- l. 193. A more detailed explanation is needed about why some multi-state characters were ordered. Ordering character states following an evolutionary hypothesis is problematic because it implies circular reasoning. As it is explained in l.962 “While a phylogenetic analysis may be a test of inferences, it is not a test that can be based on inferences”.

---

## Round 0.2 · Major Revisions

Two of the previous reviewers have commented on the manuscript and agree it has improved markedly. One just has quite minor comments. Another's comments are also mostly minor but a revision with partial uncertainty changed to polymorphism in the TNT analysis is recommended. Overall revisions are moderate, so there should not need to be more than re-review by the 1 reviewer if things proceed as expected. Well done.

·

Basic reporting

The paper is now mostly fine. I found very minor problems that could be fixed at the page proof stage (see below, section 4, Additional comments).

Experimental design

Fine.

Validity of the findings

Good.

Additional comments

I think that the author has done a good job of addressing my comments, so I feel that the paper is ready for publication, or could be subject to very minor changes, which I list below.

In this report, I refer to the line numbers in the pdf file.

Line 46: Replace “essentiality » by « great importance” because some migh confuse the former with relating to essentialism.

Line 58: I would replace “agree” in “and therein agree is well-designed » by “ assume” or both: “agree or assume” because as this draft shows, this may be more frequently an assumption than an explicit agreement made after checking data quality.

Lines 71-72: Thank you for specifying what you mean by “amphibian”, but a slight rewording is required. Please change:
Temnospondyli, often referred to as ‘amphibians’ (nonamniote tetrapods in a broad historical sense and as the putative amphibian stem-group in more recent work),
Into:
Temnospondyli, often referred to as ‘amphibians’ (nonamniote tetrapods in a broad historical sense and as the putative amphibian stem-group in several recent works),

The current wording has two problems: the use of the singular implies a single work supports the TH, which even the author surely does not believe, and second, the “more” suggests that alternatives (PH, LH) have been abandoned, which is also wrong.

These very minor changes could be made in the page proofs.

·

Basic reporting

Ready for publication.

Experimental design

Almost ready for publication: TNT analyses should be repeated with partial uncertainty changed to polymorphism (if that is even necessary) rather than to full uncertainty, which has apparently distorted the results a little.

Validity of the findings

Almost ready for publication.

Additional comments

The manuscript is approaching perfection! Here are my remaining comments, of which only the first three, arguably four, concern substance. I ask for "major revisions" only because I would like to take one last brief look at the manuscript before it is accepted; the revisions I suggest below are quite minor.

In lines 267–270 and 1251–1254, you state that because TNT cannot apparently handle partial uncertainty, you changed those scores to full uncertainty for the TNT analyses. That must be the reason why branch-and-bound in PAUP* and implicit enumeration in TNT didn’t always give the same results: the matrices were different! TNT can handle polymorphism, and polymorphism is computationally exactly the same as partial uncertainty (because ancestors cannot be reconstructed as polymorphic); the only difference is that every polymorphism adds one step to the tree after its topology has been calculated, while partial uncertainty doesn’t (i.e., polymorphism is treated as requiring a step inside the OTUs that have it, while OTUs with partial uncertainty are simply optimized as having one particular state). That is why PAUP* lets you choose whether to treat both polymorphism and partial uncertainty as polymorphism (pset mstaxa=polymorph) or as partial uncertainty (pset mstaxa=uncertain) or to treat them differently (pset mstaxa=variable). I must ask you to repeat all TNT analyses; I predict the resulting topologies and numbers of MPTs will be identical to those found by PAUP*. (I suppose it’s possible, however, that Dilkes [2020] really didn’t let his PAUP* analyses run for long enough to find any MPTs.) If TNT gives you an error message, change all partial uncertainty to polymorphism by hand, but I think TNT will read the partial uncertainties just fine and simply treat them the same way as polymorphism.
1250: Does that mean implicit enumeration only works for binary characters???
1701–1702: Is stemward slippage of small/immature specimens really expected when the clade found next to Dissorophidae consists almost only of large-bodied trematopids? Shouldn’t the small/immature dissorophids instead cluster as far away from the large trematopids as possible?
Reply to reviews: I’m not happy with the approach by Brazeau, Guillerme & Smith (2017), as I wrote in the 2019 paper (p. 21): “Too recently for us to use, Brazeau, Guillerme & Smith (2017) published a new approach to dealing with reductively coded inapplicability in phylogenetics software. We are looking forward to further developments of its implementation. However, we strongly disagree that so-called “neomorphic characters” should be scored as having the presumedly plesiomorphic state when they are inapplicable. This requires identifying the plesiomorphic state in advance, which increases the danger that the phylogenetic analysis will conform to one’s preconceptions just as much as an all-zero outgroup would. It is also much less easy than Brazeau, Guillerme & Smith (2017: 23) implied when they stated that in their analysis “every inapplicable token in each neomorphic character was replaced with the token corresponding to the presumed non-derived condition (typically ‘absent’)”—for example, our matrix contains many characters for the presence or absence of bones that are, in our taxon sample, plesiomorphically present and are apomorphically lost several times, while different taxon samples (e.g. vertebrates generally, or actinopterygians) would support the opposite polarization or none at all. Further, this method creates redundancy just like absence coding does: in the example by Brazeau, Guillerme & Smith (2017: table 1), absence of the tail unfailingly predicts absence of eyespots on the tail.”

Then there is the matter that I only just noticed Hystrix (Old World porcupines) is feminine (with species names like H. cristata Linnaeus, 1758). Therefore, so is Platyhystrix, so its species name is automatically P. rugosa. No nomenclatural act is necessary. ICZN Article 34.2 has already taken care of it: “The ending of a Latin or latinized adjectival or participial species-group name must agree in gender with the generic name with which it is at any time combined [Art. 31.2]; if the gender ending is incorrect it must be changed accordingly (the author and date of the name remain unchanged [Art. 50.3.2]).” In other words, as soon as Williston (1911) transferred Ctenosaurus rugosus Case, 1910, to his new genus Platyhystrix, it became Platyhystrix rugosa (Case, 1910), regardless of whether Williston was aware of that fact and regardless of whether that form has ever actually appeared in print.

Lines 89–90: “in phylogenetics”, as an insertion into the sentence, needs either to be surrounded by commas at both ends or none at all.
208: The coauthors and year of Zaragüeta i Bagils are missing.
250: A citation for this overestimation would be good. The phenomenon is widely known, but I don’t personally know where it has been studied.
401: Remove the apostrophe.
429: Replace “Cacops” by “Cacopus”, and replace the parenthesis by “(which is ‘Cacopod’, not ‘Cacop’)”.
430: “per the same article”, as an insertion into the sentence, needs either to be surrounded by commas at both ends or none at all. “1994”, as an insertion into the sentence, likewise needs to be surrounded by commas on both sides.
484: Replace “for” by “form”.
536: Remove the comma after “lucius”.
538: “earliest-diverging” needs its hyphen because Conjunctio is the cacopine that diverges earliest, not the earliest one of diverging cacopines.
556–557 and fig. 10A, B: No, Xerodromes has a node-based definition (as indicated by “least inclusive clade” in the original wording; the external specifiers are part of a qualifying clause), so Perryella, not Micromelerpeton, is the sister-group of Xerodromes in these trees.
609–610, fig. 6A and fig. 13A: Olsoniformes is misplaced; having a node-based definition (as for Xerodromes, the external specifiers are part of a qualifying clause), its first member is one node more highly nested, so that Iratusaurus, Palodromeus and Reiszerpeton are not members in fig. 6A, and Reiszerpeton is the sister-group of Olsoniformes in lines 609–610 and in fig. 13A. In fig. 6A it might also be a good idea, as a matter of visual presentation, to move Kamacops to the other side of Dissorophidae, so that all the single-OTU members of the dissorophid polytomy are shown on the same side.
640: Remove “were”, or replace it by “that were” or “which were”.
649-650: see 538.
698–699, fig. 18C: That’s true for nominal dissorophids, so inserting “the nominal” at the beginning of line 699 would repair the text. As defined, however, Dissorophidae would include Ecolsonia, Mordex, Rotaryus, Tambachia, Anconastes and Actiobates, so the dot that marks the basal node of Dissorophidae in fig. 18C must wander rootward accordingly. Conversely, Mattauschia is the sister-group of Olsoniformes in that strange tree.
732–733, 819, 833, 837: see 538.
829: Replace “slightly” by “slight”.
1100: “thereby”, as an insertion into the sentence, needs either to be surrounded by commas at both ends (for emphasis) or none at all (which I would personally prefer here).
1227: Uppercase R in PAUPRat, which I recently used for the first time (currently in review); it is not part of PAUP*, not widely known, and appears not to have been used ever before in phylogenetic analyses of early tetrapodomorphs. (Ruta, Coates & Quicke [2003] and Ruta & Coates [2007] did use the parsimony ratchet, but through Quicke’s unpublished, undocumented implementation rather than through PAUPRat.)
1252: Insert “not” before “others”, or remove the entire parenthesis.
1319: Replace “poor” by “poorly”.
1361: Replace “the paraphyletic Olsoniformes of” by “why Olsoniformes was not found in” – Olsoniformes has a definition with a qualifying clause, so it is either a clade or does not exist; it can’t be paraphyletic.
1379: Similarly, Trematopidae is a clade by definition, so what that tree really does is to restrict Trematopidae to Acheloma + Phonerpeton; I suggest replacing “Trematopidae” by “the nominal trematopids” or similar.
1396–1397: Likewise, Kamacopini is a clade by definition, the question is just whether it contains anything other than Kamacops. I would simply delete “and whether it is even a clade”, because the problem of its composition is already mentioned.
1526: “however”, as an insertion into the sentence, needs to be surrounded by commas from both sides.
1544: “Therein” doesn’t seem to make sense here (it triggers the question “in what?”); I recommend “Therefore”.
1551: I actually recommend “morphospecies” rather than “species” to avoid all issues with species concepts.
1581–1582: “illustrations” or “represents”?
1773: Replace “of” by “after”.
Fig. 6B, 22G: Replace “neglecta” by “neglectus”.
Fig. 14A: Remove the dot for Xerodromes (as the text correctly states).

Finally, let me strenuously disagree with Reviewer 4 on one point: I really hope it was never common practice to score assumed/inferred character states in phylogenetic analyses of any taxon. In the fields I know, it has not been usual in about 30 years, if ever. In the mid-late 90s, Paul Sereno once said (or so I was told) about his analyses of the phylogeny of various dinosaur clades that once you’re experienced enough, “you know what the tree is going to look like, and then you code accordingly.” Indeed, his analyses, unlike everyone else’s, had CIs above 0.8, sometimes above 0.9; given the size of his matrices, such CIs are simply impossible to reach in any honest way. He was at least cherry-picking his data and thought that was a good thing. Even his own slightly later analyses, however, had much more realistic CIs (and different topologies, too…). He had evidently learned that if you score your OTUs after their presumed close relatives, you’re committing circular logic. The analysis is supposed to tell you if they really are close relatives. It is not supposed to merely illustrate your opinions (as was done in some works in the late 80s and early 90s), it’s supposed to give you an independent result based on an independent assessment of the evidence – independent of you, that is. If you assume a tree a priori and invent scores for missing data to fit the tree, your analysis will give the tree you assumed back to you, and you’ll have no way of telling if that’s accurate – which it may well be! – or just another case of “garbage in, garbage out”. Do not score states that aren’t preserved in the actual material, no matter how reasonably you assume them. Phylogenetic analysis is supposed to be science, not circular logic. Or as the manuscript says in lines 1001–1002: “a phylogenetic analysis should inform taxonomy, rather than vice versa.” – The manuscript already makes clear, e.g. in lines 353–355, that the malpractice wasn’t limited to Schoch for dissorophids, but that e.g. Holmes, Berman & Anderson (2013) committed it as well. – Too bad the archives of the Dinosaur Mailing List are currently down. Some 20 to 25 years ago, the question actually came up if Tyrannosaurus should be scored as laying eggs in a phylogenetic analysis. None less than Thomas Holtz, a pioneer of theropod phylogeny, spoke up and said no: there is no fossil evidence on the reproduction of Tyrannosaurus specifically, so the correct score for Tyrannosaurus is a question mark. As a matter of fact, we don’t know if Tyrannosaurus had any unexpected autapomorphies in this respect, no matter how unlikely it may seem; and before the analysis is done, we also don’t really know if reproduction evolved in a more complex way than we think. That post might have been citable, should a citation really be felt to be necessary.

On ordering, I agree with Reviewer 4 that ordering based on an evolutionary hypothesis runs the risk of circularity and should accordingly be avoided; but that is not the same as ordering characters that represent a morphocline, because such characters can be recognized in the absence of any phylogenetic/evolutionary hypothesis. Moreover, the assumption that it is easier to change from one state to a close state than to a distant state has already been used to divide the character into discrete states in the first place, so it would be inconsistent not to use it again. Citations for this are in the manuscript and in my & Laurin’s 2019 paper.

David Marjanović

---

## Round 0.3 · Minor Revisions

The reviewer has some minor textual suggestions for changes, without need for further review; these seem easily done. Thank you!

·

Basic reporting

no comment

Experimental design

no comment

Validity of the findings

no comment

Additional comments

The finding that implicit enumeration in TNT and branch-and-bound in PAUP* produce different results seems very important, but also beyond the scope of the manuscript. Maybe add something, after line 1269 of the tracked-changes manuscript, to the effect that their behavior in phylogenetic inference should be studied further.

209: Replace “Rineua” by “Rineau”.
276: Close the parenthesis at the end of the sentence.
438: Insert a comma after “‘Cacopod’”.
446: I’m not sure how best to word this, but “should have been” is not the best option. Maybe “was in fact” or “was, as a matter of legal fiction,”…

That’s it – I recommend acceptance after the minor issues above are resolved.

David Marjanović

---

## Round 0.4 · accepted · Accept

I have checked the changes and the manuscript is in fine shape. Thanks for persevering-- the study has improved nicely. Congratulations!!!!